# Full-waveform tomography reveals iron spin crossover in Earth's lower mantle

Laura Cobden [1] ✉, Jingyi Zhuang [2], Wenjie Lei[2,3,8],
Renata Wentzcovitch [2,4,5,6,7] ✉, Jeannot Trampert [1] & Jeroen Tromp [3]

Three-dimensional models of Earth's seismic structure can be used to identify temperature-dependent phenomena, including mineralogical phase and spin transformations, that are obscured in 1-D spherical averages. Full-waveform tomography maps seismic wave-speeds inside the Earth in three dimensions, at a higher resolution than classical methods. By providing absolute wave speeds (rather than perturbations) and simultaneously constraining bulk and shear wave speeds over the same frequency range, it becomes feasible to distinguish variations in temperature from changes in composition or spin state. We present a quantitative joint interpretation of bulk and shear wave speeds in the lower mantle, using a recently published full-waveform tomography model. At all depths the diversity of wave speeds cannot be explained by an isochemical mantle. Between 1000 and 2500 km depth, hypothetical mantle models containing an electronic spin crossover in ferropericlase provide a significantly better fit to the wave-speed distributions, as well as more realistic temperatures and silica contents, than models without a spin crossover. Below 2500 km, wave speed distributions are explained by an enrichment in silica towards the core-mantle boundary. This silica enrichment may represent the fractionated remains of an ancient basal magma ocean.

Seismic tomography provides maps of the wave-speed structure inside the Earth's mantle but interpreting these maps in terms of dynamically relevant parameters such as temperature and mineralogy is a formidable task[1,2]. Different thermochemical parameters can have opposing effects and hence "trade off" with each other to produce a given wave-speed value[3]. Breaking this trade-off requires a joint interpretation of multiple observables, such as compressional (*P*) and shear (*S*) wave speeds together, or wave speeds with density.

While there are many different *P* and *S* wave-speed models for the lower mantle, most were obtained independently with different data sets and at different seismic frequencies, rendering a joint interpretation ineffectual[4,5]. Additionally, the majority of these models were derived using classical methods which utilize only a fraction of the information available in a seismogram, and which do not capture complex wave phenomena such as diffraction. 3-D wave speeds in such tomographic models are usually expressed as linear perturbations from a reference model, rather than absolute values. In the lower mantle, with the exception of the lowermost 300-400 km (D″ layer), these perturbations are small – mostly less than 1-2%. This further obfuscates a quantitative interpretation because many different combinations of thermal or compositional changes – as well as errors – can produce the observed signals. Furthermore, damping and regularization underestimate the true amplitudes of wave speed variations[6].

Tomography models derived via full-waveform inversion, that are based on fitting whole seismograms and in which the physics of wave

[1]Department of Earth Sciences, Utrecht University, 3584 CB Utrecht, Utrecht, The Netherlands. [2]Department of Earth and Environmental Sciences, Columbia University, New York, NY 10027, USA. [3]Department of Geosciences, Princeton University, Princeton, NJ 08544, USA. [4]Department of Applied Physics and Applied Mathematics, Columbia University, New York, NY 10027, USA. [5]Lamont Doherty Earth Observatory, Palisades, NY 10964, USA. [6]Data Science Institute, Columbia University, New York, NY 10027, USA. [7]Center for Computational Quantum Physics, Flatiron Institute, New York, NY 10010, USA. [8]Present address: Google Inc., Mountain View, CA, USA. ✉e-mail: l.j.cobden@uu.nl; rmw2150@columbia.edu

propagation is accurately incorporated, provide images of the Earth's interior that are both sharper in resolution and show larger amplitude variations[6]. Additionally, the iterative, non-linear inversion procedure directly delivers absolute wave speeds, significantly improving the constraints that can be placed on the underlying physical properties[7].

While the theoretical background for full-waveform inversion was developed almost 40 years ago (e.g.,[8]), performing such calculations on a global scale has only just become computationally feasible[9]. We present a physical interpretation of a recently published global full-waveform tomography model[10], GLAD-M25, between 1000 and 2800 km depth. GLAD-M25 constrains bulk and shear wave speeds simultaneously using the same data and over the same range of seismic frequencies (everything in the range 17–150 s) and has excellent data coverage for both P and S waves traversing the lower mantle. The model resolution has been determined using point spread functions (see Supplementary Material in[10]), which demonstrate that the P and S wave speeds are well-resolved in the mid-mantle. Furthermore, GLAD-M25 is based on frequency-dependent travel time measurements and not amplitude measurements such that the tomography model is not biased by differential amplitudes between P and S arrivals. Remarkably, using phase and travel time measurements alone, GLAD-M25 reconstructs the amplitudes of the waveforms almost completely. These factors make a joint interpretation of the P and S wave speeds meaningful.

The bulk wave speed, $V_\Phi = \sqrt{K/\rho}$, is obtained through a simple combination of the compressional ($V_P = \sqrt{(K+(4/3)G)/\rho}$, and shear ($V_S = \sqrt{G/\rho}$) wave speeds, where $K$ is the bulk modulus (incompressibility) of the material, $G$ is the shear modulus (rigidity) and $\rho$ is the density, i.e. $V_\Phi^2 = V_P^2 - (4/3)V_S^2$. Interpretation of P-wave speed directly is challenging because it depends on both the bulk and shear moduli that are differentially influenced by mineral physics processes. Hence, the separation of the wave speeds into bulk and shear components facilitates interpretation.

Here, we study the frequency distributions (i.e., histograms) of shear and bulk wave speeds as a function of depth in GLAD-M25. By forward modeling the elastic properties of candidate mantle models via mineral physics, these histograms provide information on the ranges of temperature and composition at different depths in the mantle, as well as any first-order mineralogical transformations that influence seismic properties[11]. The power of working with globally-compiled histograms rather than local wavespeeds (i.e. wave-speeds associated with a specific latitude and longitude) is that the effect of uncertainties in either the mineral physics or tomography is much smaller. Additionally, as opposed to 1D spherical averages[12,13], these distributions display more clearly the fingerprint of the spin crossover within any region containing ferropericlase. This is a distinct depth-dependent temperature-induced change in the seismic velocities, with $dVs/dT < 0$ while $dV_\Phi/dT \geq 0$.

We create our candidate mantle models by selecting temperatures and compositions randomly from pre-defined ranges (the Prior) in a Monte-Carlo procedure. For each selected temperature and composition, we calculate the equilibrium mineral phase assemblage via a Gibbs energy minimization and use equation-of-state modeling to compute the bulk and shear wave speeds of the assemblage. Wave speeds are adjusted for temperature-dependent anelasticity, although the effect of this correction on the wave speeds at body-wave frequencies is very small (see Methods for further details). We repeat this procedure hundreds of thousands of times for a given prior.

## Results

### Effect of fixed vs variable composition in the lower mantle
We consider three different priors for the lower mantle composition (Supplementary Fig. S1). In the first, all models have a pyrolite composition. Pyrolite[14] is the hypothetical source material for mid-ocean ridge basalts (MORB), and therefore geodynamic models of mantle

convection, as well as mineral physics calculations, often begin with the assumption that this is the average bulk composition of the lower mantle. The exact definition of pyrolite varies between petrological studies, so we allow minor changes in composition between pyrolite models to accommodate this uncertainty.

In our second prior, we allow very broad variations in chemistry; extending continuously from the ranges seen in mantle xenoliths[15] up to the values seen in MORBs e.g.[16] and chondritic Earth models[17]. While this gives a lot of freedom in compositional possibilities for the mantle, it also includes many intermediate compositions between pyrolite and MORB that are not realistic, and it is furthermore questionable if subducted oceanic crust can be resolved at the length-scales of seismic tomography. Therefore, in our third prior we again vary the chemistry, covering the full variability seen in xenoliths and beyond, but the ranges are more restricted such that MORB-like models are excluded.

A simple, effective method to assess the relative feasibility of the three priors is to look at scatter plots of bulk versus shear wave speed at different depths. An example is shown in Fig. 1 at depth intervals of 300 km. In the pyrolite models (Fig. 1a), wave-speed variations follow a narrow diagonal trend due to temperature variations, and clearly, these models cannot capture the diversity of the bulk and shear wave speeds in GLAD-M25 simultaneously at any depth (although they fit the ranges of either the bulk or the shear wave speeds in isolation). Assuming a different fixed composition than pyrolite would shift the clouds of the models without expanding their scatter. This gives a strong indication that variations in chemistry are required to explain seismic wave speeds in the lower mantle.

With broad variations in chemistry it is possible to generate bulk and shear wave speeds which match the diversity of wave speeds seen in GLAD-M25 (Fig. 1b). With restricted variations in chemistry (Fig. 1c), this is possible at the top of the lower mantle, but with increasing depth the overlap between the synthetic models and GLAD-M25 decreases, before improving again in the D″ region.

### Effect of an iron spin crossover
In order to improve the fit in the mid-mantle with the "restricted" prior, we require a mechanism that reduces $V_\Phi$ relative to $V_S$. Both experiments and theoretical calculations have predicted that $Fe^{2+}$ in (Mg,Fe)O (ferropericlase) is susceptible to a spin state change[18,19]. The spin state refers to the occupancy of the 3d orbitals in iron: in ferrous iron in the high-spin (HS) state, four electrons occupy unpaired orbitals and two are paired; in the low-spin (LS) state, all six electrons are paired, thus occupying three orbitals only. At low temperatures the crossover takes place relatively abruptly, but along a lower mantle geotherm, the HS-LS crossover takes place over a broad pressure/depth range[20,21] (Fig. S18). This is known as the spin crossover region, in which both spin states coexist – i.e. the "mixed spin" (MS) state. Owing to theoretical approximations and experimental limitations, there is still some uncertainty on the exact depth onset and thickness of the iron spin crossover (ISC) region at mantle temperatures.

The ISC in ferropericlase is associated with a significant softening of the bulk modulus[22,23], and smaller changes in the shear modulus and density (Fig. S5). This is in line with what our variable-composition models require to better fit seismic observations (Fig. 1c). However, because of the gradual and smooth nature of the ISC, and the fact that ferropericlase is expected to constitute not more than ~15-20 vol% of the bulk mineralogy, its effect on seismic properties may be subtle. The spin crossover cannot be readily discerned in spherically-symmetric 1-D reference models such as PREM[12] and AK135[13]. This is unsurprising since it would likely manifest as a change in velocity gradient with depth rather than a sharp discontinuity[24]. The gradients in these 1-D reference models are pre-determined by the parameterization choices during the inversion procedure[25]. Interestingly, the 1-D radial average for $V_P/V_S$ in GLAD-M25 does display a small decrease in the mid-mantle which may be compatible with a spin crossover (see Figure A1

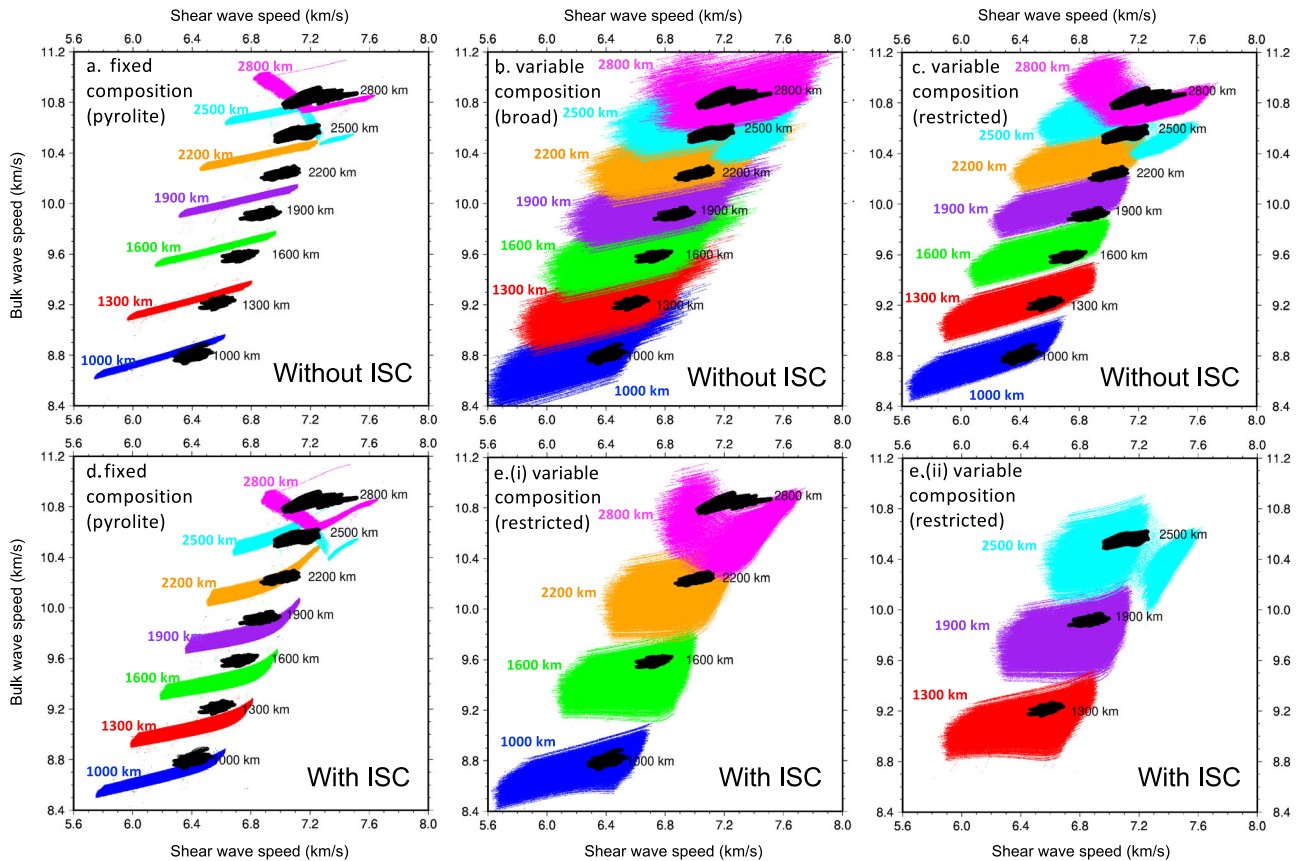

**Fig. 1 | Scatter plots showing ranges of bulk and shear wave speed in seismic tomography (GLAD-M25) versus different thermochemical priors.** Black clouds are for GLAD-M25, where wave speeds have been specified at every 1 degree latitude and longitude and comprise 65,341 data points (i.e., 181 lat × 361 lon). Colored clouds are for thermochemical models: 330,000 pyrolite models and 750,000 variable composition models. Prior models are selected at random from the ranges shown in Fig. S1. Top row: models without a correction for iron spin crossover (ISC) in ferropericlase. Bottom row: models with a correction for ISC in ferropericlase (**a**, **d**) pyrolite (Prior 1); (**b**) broad variable composition (Prior 2), (**c**, **e**) restricted variable composition (Prior 3). E is split into two plots (i and ii) in order to visualize overlapping ranges. **d**, **e** were calculated assuming non-ideal mixing between the high spin (HS) and low spin (LS) states. Plot (d) is without magnetic effects while Plot (e) is with magnetic effects (see "Methods" section and Fig. S3 for further information).

in Lei et al.[10]). We are cautious to make such an interpretation, as 3-D variations in physical structure may produce a 1-D average structure which has no physical basis (and whose interpretation is non-unique). The ISC, whose depth onset depends on both temperature and composition, is expected to have significant 3-D variations in a convecting mantle, and 1-D spherical averages are particularly biased by phenomena which exhibit strong lateral variations[26]. We therefore study the full distributions of the wave speeds provided by seismic tomography, and not just the mean value reflected by the 1-D average.

Recently, evidence for a spin crossover has been suggested on the basis of differential abundances of "fast" and "slow" wave speeds between $P$ and $S$ wave speed tomography models[27]. However, the tomography models used in[27] were derived at different seismic frequencies and resolution, with different methods and datasets and are not fully consistent with each other. Conclusions about the presence of the ISC were based on individual models but were best discerned through a "vote map" technique that extracts the most robust and common qualitative patterns in these models. Here, we apply a fully quantitative approach with a tomography model in which the $P$ and $S$ wave speeds are demonstrably consistent with each other.

The effect of the ISC on seismic wave speeds were recalculated using a novel non-ideal HS-LS mixing ab initio model for ferropericlase (see Methods for details). The non-ideal HS-LS mixing broadens considerably the ISC depth range (Fig. S18). We investigated the effect of two approximations on the ISC pressure/temperature range: ideal vs. non-ideal HS-LS mixing and magnetic entropy. Using the newly

calculated velocities we correct the bulk modulus, shear modulus, and density of our prior models accordingly. Inclusion of the ISC effect on bulk and shear moduli expands the ranges of scatter plots (Fig. 1) significantly. For fixed composition (pyrolite) models, the scatter of bulk and shear wave speeds still does not overlap with GLAD-M25 (Fig. 1d), but for models with variable composition (Prior 3) it becomes possible to fit GLAD-M25 everywhere above D" (Fig. 1e).

We can quantify the relative fit of the models with and without the ISC by applying a Metropolis-Hastings algorithm[28]. This algorithm compares histograms between a target distribution (GLAD-M25) and a prior distribution (the minerological models). The concept is illustrated in Fig. S6. The algorithm selects a subset of models from the prior based on the relative heights of the bars in the two histograms, to replicate the target distribution as close as possible. The algorithm considers each bar of the histogram in turn. Where the frequency of the prior is bigger than the target frequency for that bar, the algorithm discards a fraction of the mineralogical models in proportion to the relative heights of the bars and using a random number generator to select which particular models are discarded. Where the frequency of the prior is less than the target frequency, all models are retained. New histograms are constructed for the retained models, and this procedure is iterated until the best fit between the retained models and the target is obtained. With ~10⁶ models in the prior, the algorithm usually converges within 10-20 iterations, but the exact number depends on how similar the two histograms are initially.

We quantify the fit of a given set of retained models (the posterior) to the target distribution ($V_\Phi$ or $V_S$ of GLAD-M25) as follows: For the $i$'th bar of the histogram, we define $f_{min}$ as whichever is the lower value out of: the frequency of the posterior ($f_p$) and the frequency of the target ($f_t$). Since both distributions are normalized to 100%, the fit is then given by the sum of $f_{min}$ over all values of $i$. In other words:

$$fit\ (\%) = \sum_i \min(f_p, f_t)$$

We use this approach because it is simple and illustrates the degree of fit clearly.

In our application of the Metropolis-Hastings algorithm, we fit the frequency distributions of shear and bulk wave speeds simultaneously, at a given depth. For Prior 3 (variable composition, restricted ranges), inclusion of the ISC improves the fit substantially between ~1800 and 2500 km (Fig. 2), giving us an indication of the depth range in the mantle where the ISC is most prevalent. This improvement holds for all four approximations made in the ferropericlase ISC modeling and clearly originates in the presence of the ISC (Fig. S20). In particular, the ISC models with magnetic entropic effects (MEE) offer the best fit. These models broaden the ISC along the mantle geotherm. The non-ideal SS model with MEE gives the broadest ISC range in the mantle and fits mantle seismic wave speeds best. In the D″ region, models with an ISC correction fit the same as, or worse than, models without this correction.

By studying the physical properties of the "best fitting" models (i.e., those retained by the Metropolis-Hastings algorithm), we can further constrain the frequency distributions of temperature, bulk composition, and mineralogy which can reconstruct the wave speed distributions of GLAD-M25. The key findings are shown in Fig. 3. Although "broad" chemical variations can fit seismic observations equally well with or without an ISC (Prior 2, Fig. S7), models without an ISC are very cold (~800 K below a standard mantle adiabat) and they

compensate for not having a spin crossover with a major depletion in $SiO_2$ in the mid-lower mantle (this manifests as a depletion in bridgmanite). The values of $SiO_2$ (<36 wt%) are much lower than any model proposed for the bulk mantle composition on the basis of petrological or cosmochemical arguments[14,17,29–31] and extend over a depth range of several hundred kilometers, which is volumetrically significant when integrated over the whole mantle. Including the effects of an ISC results in mantle temperatures and silica contents which are geodynamically and geochemically more plausible, for both broad and restricted variations in chemistry.

In the D″ region, bulk wave speed increases at a faster rate than in the overlying mantle. Our thermochemical models cater for this with an enrichment in $SiO_2$ towards the core-mantle boundary (Fig. 3), regardless of the prior ranges in chemistry or whether an ISC correction is applied. Models selected from a broad prior (Prior 2) can accommodate a greater enrichment in $SiO_2$, and these models also fit the seismic data better than the models based on the narrow prior (Prior 3) (compare Figs. 2 and S7). In the absence of sufficiently $SiO_2$-rich models, the models taken from Prior 3 compensate by reducing the iron content (Fig. S9) but this then provides a less optimal fit to shear and bulk properties simultaneously.

Reducing the iron content also reduces the density. Although our results are based on fitting wave speeds, density is implicitly calculated in our simulations, and we can compare the density distributions of our thermochemical models with PREM (Fig. S14). We find that above D″, both Prior 2 and Prior 3 are compatible with PREM. However, in the D″ region, the more Si-enriched models (from Prior 2) are more similar to PREM than the restricted-$SiO_2$ models (from Prior 3).

## Discussion

While the existence of a widespread ISC in Earth's lower mantle was hypothesized over 30 years ago[18], a seismic signature was not anticipated until recently[23,24,32]. Ascertaining the presence of the crossover is important because the redistribution of electrons in $Fe^{2+}$ alters the thermal and electrical conductivity of ferropericlase, as well as the viscosity[19,33,34], which in turn may influence the convection dynamics inside the Earth, in particular the stability of chemical piles[35]. Previous studies e.g.,[27] have been based on demonstrating consistency between theoretical predictions of the ISC and seismic observations. In this study, we instead quantitatively compare the fit of mantle models with and without an ISC. By modeling absolute wave speeds (rather than perturbations from a 1-D average), we can constrain absolute temperatures and compositions, which in turn allows us to distinguish which scenarios are (im)plausible. We demonstrate that including the elastic effects of the ISC in ferropericlase fits seismic tomography better, and that alternative explanations for the observations, namely a change in bulk chemistry by $SiO_2$ depletion, are unfeasible. Using bulk wave speed rather than $P$-wave speed enhances the signal of the ISC[22,32,36] (Fig. S15).

We have exhaustively addressed the uncertainty in our ISC calculations by including four different theoretical models. Uncertainties in other mineral elastic parameters are unlikely to affect our interpretation because we expect that errors in the predicted elastic properties would either shift the wave speeds by the same amount at all depths, or would systematically increase with depth towards the core-mantle boundary. However, the signal that we see in the seismic data is a distinct "anomaly" that is restricted to a few hundred kilometers in the mid-mantle. Our findings also cannot be explained by anelasticity effects, as anelasticity reduces the shear wave speed relative to the bulk wave speed, whereas our data require the opposite, i.e. a reduction the bulk wave speed relative to the shear wave speed.

Ferric iron ($Fe^{3+}$) in bridgmanite may also experience a HS-LS crossover under lower mantle conditions and is also associated with a reduction in bulk modulus[37]. This effect is however expected to be smaller than that in ferropericlase owing to the smaller concentration

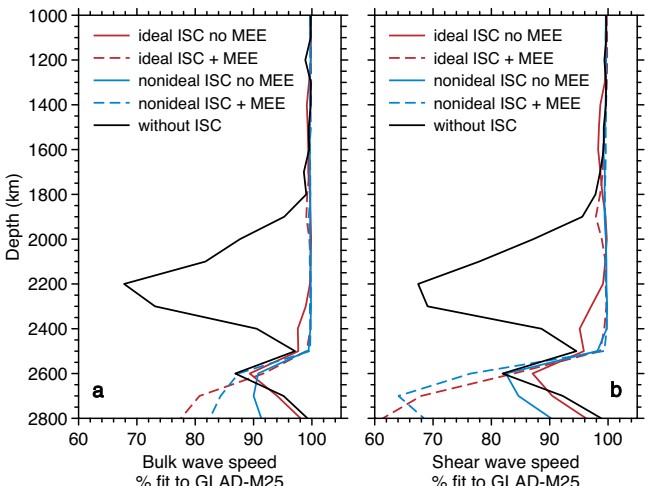

**Fig. 2 | Percent fit of thermochemical models to the bulk and shear wave speed distributions in GLAD-M25 as a function of depth.** Percent fit is defined as the degree of overlap between theoretical wavespeed frequency distributions and observed wavespeed frequency distributions (example frequency distributions are shown in Fig. S6) (**a**) fit to bulk wave speed distributions and (**b**) fit to shear wave speed distributions. Solid black lines are models without an iron spin crossover (ISC) in ferropericlase. Colored lines show the effect of the ISC in ferropericlase using four different approximate models: ideal (red) vs. non-ideal (blue) HS-LS mixing combined with or without magnetic entropic effects (MEE) (see details in Methods). The results plotted here are for thermochemical models with restricted variations in chemistry (Prior 3; Fig. S1). Including the ISC significantly improves the fit between ~1800–2500 km depth.

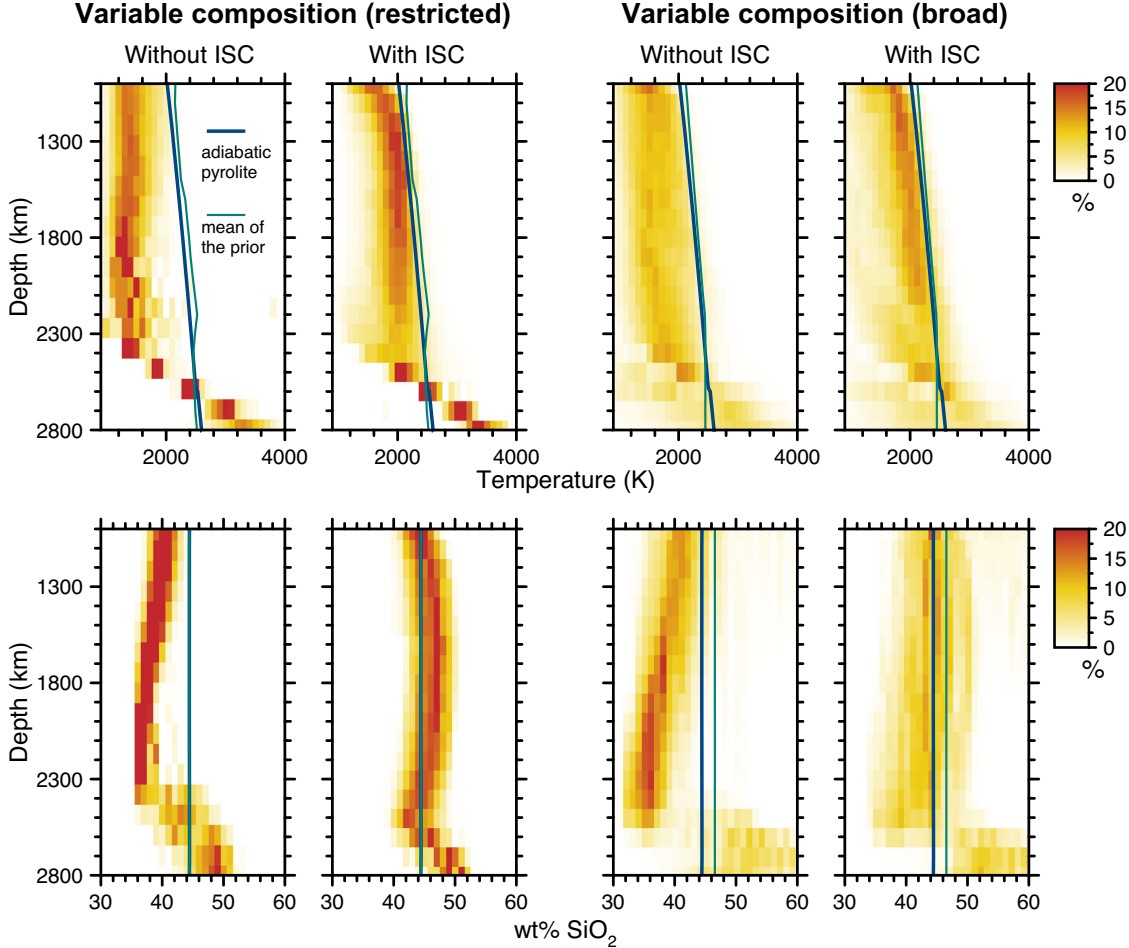

**Fig. 3 | Density plots showing the distributions of temperature (top row) and silica content (bottom row) as a function of depth for the subset of thermochemical models which best fit GLAD-M25.** The darker the color, the higher the density of models (see legends). For reference, the thermal structure of an adiabat for pyrolite with a potential temperature 1573 K is also shown (blue line) together with the mean of the prior (green line). Without a correction to the wave speeds for an iron spin crossover (ISC) in ferropericlase, models are unrealistically cold and have very low SiO₂ in the mid-mantle (especially between ~1800–2400 km depth).

All models show an enrichment in SiO₂ in the D″ region, increasing towards the core-mantle boundary. Left panels are for restricted variations in chemistry (Prior 3) and right panels are for broad variations in chemistry (Prior 2). See Fig. S1 for chemical ranges of the different priors. We used results from Model-4 for the ISC (non-ideal HS-LS mixing including full magnetic entropic effects). The other three ISC models are shown in the Supplementary Materials, Fig. S8. Results for other chemical/mineralogical parameters are shown in Figs. S9–S13.

of $Fe^{3+}$ in the average mantle, and is suppressed by the presence of aluminum[38,39]. We can fit seismic wave speeds completely between 1000 and 2500 km depth by considering the ISC in ferropericlase alone, but it is possible that a similar ISC in bridgmanite may contribute to the observed signal, especially in the large low shear velocity provinces (LLSVPs)[40].

Seismic tomography models depicting slab-like features traversing the whole mantle are often viewed as evidence for a chemically homogenous, well-mixed mantle. Our best-fitting mantle models require chemical heterogeneity at all depths in the lower mantle, and especially below 2500 km. A strong enrichment in silica in the D″ region may represent fractionated remnants of an ancient magma ocean[41] or MORB accumulation that are largely stable through geological time[42]. These Si-rich domains could reconcile the discrepancy in Mg/Si ratio between upper mantle rocks and chondritic meteorites[17]. Future analysis of the 3-D wave-speeds geographically may reveal how such enrichment is distributed laterally. Additionally, ongoing studies in mineral physics will enable the inclusion of the effects of ferric iron[40] and calcium-rich bridgmanite[43], as well as further constraints on the stability and elastic properties of post-perovskite, which may influence the

seismic properties in this region and subsequently modify the inferred chemical composition.

## Methods

### Synthetic thermochemical models

**Building the prior.** Bulk compositions are defined in terms of the NCFMAS system (i.e., the 6 major oxides in terrestrial rocks), and models are drawn at random from three different prior distributions (Fig. S1). For each prior, we first prescribe a maximum and minimum value for the 6 oxides (Table S1), then for each oxide we select models randomly from a uniform distribution between these limits. Next, since the total weight percent of all six oxides must sum to 100%, we normalize the proportions of the six oxides to 100. For our variable composition models (Priors 2 and 3), this results in prior distributions which are non-uniform but whose peak ranges are similar to the distributions seen in xenoliths[15], and whose tails accommodate more extreme rock types. Prior to normalization, models are selected independently for each oxide, but normalization inevitably introduces a degree of anti-correlation between the most abundant oxides (MgO and SiO₂) since the total amount cannot exceed 100%. The other components are uncorrelated (Figure S2).

For Prior 1 (pyrolite), minor variations in chemistry are included to account for variability in the formal definition of pyrolite, based on refs. [14,29,31,44–48]. These minor variations also serve as a buffer for seismic uncertainties in GLAD-M25. We generate 1100 random pyrolite compositions. For Priors 2 and 3 (variable composition), we generate 2500 random compositions each, based on xenoliths[15], MORBs[49] and lower mantle models derived from chondritic and solar abundances[29,30].

We analyze GLAD-M25 in depth intervals of 100 km, from 1000 km to 2800 km depth, i.e. 19 depths in total. We convert these depths into corresponding pressures using the depth-to-pressure calibration of PREM[12]. At each of these pressures, we pick 300 temperatures at random from a uniform distribution, with a minimum temperature of 900 K and a maximum temperature at the solidus of $MgSiO_3$[50]. Hence, for Prior 1, this gives a total of 330,000 thermochemical models at each depth, and for each of Priors 2 and 3, this leads to a maximum of 750,000 models at each depth. The ranges of the priors determine the ranges of seismic properties (Figs. 1, S3, and S4), while the number of models per Prior determines how densely those ranges are sampled (e.g., compare the density of points for Priors 2 and 3 in Fig. S2). It is important that there are sufficient models available for the Metropolis-Hastings algorithm to converge, but more models are not necessary.

**Calculation of seismic properties.** For each of our thermochemical models, corresponding phase relations and isotropic elastic properties are calculated using *Perple_X* thermodynamic modeling software[51] together with the elastic parameters, equation of state, and solid solution model of refs. [52,53]. Occasionally with an extreme composition or at very low temperature, models are thermodynamically unstable and are discarded from the dataset. The number of remaining models after thermodynamically unstable ones have been discarded is shown in Table S2.

We first output the bulk and shear wave speed of the bulk mineral assemblage calculated directly with *Perple_X*. These wave speeds represent a thermochemical model without an iron spin crossover (ISC), and are the models plotted in Fig. 1a-c. We then adjust the wave speeds to include the effect of an ISC in ferropericlase. We consider four different theoretical approximations for the ISC (described in Section II). We compute the change in bulk modulus, shear modulus, and density as a function of temperature and iron content, relative to the high-spin (HS) state which is implicitly calculated with the database of refs. [52,53]. These properties are calibrated every 100 K in temperature and 0.01 $x_{Fe}$ in ferropericlase. For each thermochemical model, we use a 2D interpolation in Python to extract the change in elastic properties in ferropericlase at the temperature and $x_{Fe}$ value for that model. With ferropericlase's properties updated for an ISC, we re-calculate the bulk and shear wave speeds (and density) of the entire mineral assemblage using a Voigt-Reuss-Hill averaging scheme.

In this procedure, a small inconsistency is introduced, because the mineral phase relations are not modified for the ISC, while a redistribution of iron between ferropericlase and bridgmanite is expected (with more iron entering ferropericlase) e.g.[19]. Although we do not include the effect of iron partitioning across the ISC on aggregates' densities and velocities, on the basis of Birch's law, we anticipate that such effect will be relatively small. Birch's law relates the compressive velocity with the aggregate density, which depends primarily on the unchangeable aggregate composition. This topic is beyond the scope of this work and we anticipate that our conclusions will not be significantly affected by the iron partitioning effect.

While all four theoretical approximations improve the fit to GLAD-M25 in the mid-lower mantle, models that include non-ideal solid solution show more plausible thermochemical behavior after fitting to GLAD-M25 (see Figus S8-S12) and are therefore our preferred choice.

**Correction to wave speeds for intrinsic anelasticity.** As a last step, we apply a simple correction to adjust shear wave-speeds for the effect of temperature-dependent intrinsic anelasticity. We follow the procedure in[54] with the parameters given in Table S2. The anelasticity correction is performed at a period of 1 s, as we work with a version of GLAD-M25 in which the wave speeds have been shifted to 1 second period. The effect on S-wave speeds is very small (<- 0.3%).

## Iron spin-crossover modeling
**Thermodynamic modeling.** Here we model more realistically the acoustic velocities of ferropericlase (*fp*). Previously such velocities were obtained using an ideal solid-solution mixing model[55] and more approximate vibrational properties[56]. Although the importance of non-ideality on the HS-LS solid solution has been emphasized there are no predictions of this effect on seismic velocities[57]. There are two levels of modeling in the iron spin crossover (ISC) solid-solution: a) the MgO-FeO solid solution modeling is treated as a quasi-ideal solid-solution[58,59] with end-members MgO and $Mg_{(1-x)}Fe_xO$ ($x_{Fe}$ = 0.1875) in the high-spin (HS) or low-spin (LS) state. This level of modeling is equivalent to treating the solid solution as a Henryan solution, with an activity coefficient different from 1 but constant for small x; b) the $Mg_{(1-x)}Fe_x^{HS}O$ and $Mg_{(1-x)}Fe_x^{LS}O$ solution modeling with fixed x and LS fraction $n = \frac{n_{LS}}{n_{HS}+n_{LS}}$ varying in the full range 0 <n< 1. The non-ideal free energy expression is general, and its contributions were described in a recent paper[21]:

$$G_{non-ideal}(P,T,n) = G_{ideal}(P,T,n) + G_{ex}(P,T,n) \quad (1)$$

$G_{ex}(P,T,n)$ is the excess Gibbs free energy describing the non-ideal mixing and $G_{ideal}(P,T,n)$ is

$$G_{ideal}(P,T,n) = (1-n)G_{HS}(P,T) + nG_{LS}(P,T) + G_{mix}(P,T,n), \quad (2)$$

where $G_{HS/LS}(P,T)$ is the Gibbs free energy of 100% HS or LS ferropericlase, i.e.,

$$G_{HS/LS}(P,T) = F_{HS/LS}^{st+vib}(V(P),T,n) + PV_{HS/LS} + G_{mag}(P,T,n). \quad (3)$$

$P_{HS/LS} = -\frac{\partial F_{HS/LS}^{st+vib}}{\partial V}$, where $F_{HS/LS}^{st+vib}$ is the Helmholtz free energy described within the quasiharmonic approximation (QHA)[60]. $F_{HS/LS}^{st+vib}$ includes the static energy, $E^{st}(V,n)$, and the vibrational contribution, $F^{vib}(V,T,n)$:

$$F^{st+vib}(V,T,n) = E^{st}(V,n) + F^{vib}(V,T,n). \quad (4)$$

$G_{mix}(P,T,n)$ is the ideal free energy of mixing:

$$G_{mix}(P,T,n) = -TS_{conf}(T,n) = k_B T x_{Fe}[n\ln n + (1-n)\ln(1-n)], \quad (5)$$

where $S_{conf}(T,n)$ is the HS/LS configuration entropy (r.h.s. in Eq. 5) and $k_B$ is the Boltzmann constant. The magnetic contribution to the free energy in Eq. (3) is:

$$G_{mag}(P,T,n) = -TS_{mag}(T,n) = -k_B T x_{Fe}(1-n)\ln[m(2S+1)]. \quad (6)$$

where $S_{mag}(T,n)$ is the magnetic entropy (r.h.s in Eq. 6).

$G_{mag}(P,T,n)$ is non-zero for the HS state only. Equation (6) assumes no exchange interaction between iron ions (no spin-spin correlations) and corresponds to the atomic limit, where m = 3 is the minority electron orbital degeneracy in the HS state and S = 2 is the total spin of iron in the HS state. Equation (6) gives the maximum magnetic entropy allowed, which is a good approximation in the limit of small x, where Fe-Fe distances are large. For large x, Fe-Fe distances are small, and exchange interaction may induce magnetic ordering, decreasing $S_{mag}$. fp with $x_{Fe} \leq 0.2$ may still be treated well in the atomic limit (as paramagnetic impurities), but here we inspect the effect of

two limits of $S_{mag}$ on the spin-crossover: the maximum value given by Eq. (6) and the minimum value, i.e., $S_{mag} = 0$, as in a diamagnetic insulating state.

Putting all these ingredients together, we minimize $G_{non-ideal}(P,T,n)$ w.r.t $n$ to obtain the equilibrium $n(P,T)$, i.e., the solution of

$$\Delta G_{LS-HS}(P,T) + \frac{\partial G_{ex}(P,T,n)}{\partial n} + k_B T x_{Fe} \ln\left[\frac{n}{1-n}(m(2S+1))\right] = 0. \quad (7)$$

In the absence of $G_{ex}$, the solution is

$$n = \frac{1}{1 + m(2S+1)\exp\left[\frac{\Delta G_{LS-HS}(P,T)}{k_B Tx}\right]} \quad (8)$$

where $\Delta G_{LS-HS}(P,T) = G_{LS}(P,T) - G_{HS}(P,T)$. For non-vanishing $G_{ex}(P,T,n)$ (see Eq. 1), Eq. (8) needs to be solved numerically. Here we include only the static part of $G_{ex}(P,T,n)$, i.e., the temperature independent excess enthalpy,

$$G_{ex}(P,T,n) = H_{ex}(P,n) = E_{ex}^{st}(V,n) + P_{ex}(V,n)V \quad (9)$$

where $E_{ex}^{st}(V,n)$ is the excess static energy, and $P_{ex}(V,n) = -\frac{\partial E_{ex}^{st}}{\partial V}$ is the the excess static pressure, and assume $G_{ex}^{mag} = F_{ex}^{vib} = 0$. This is an excellent approximation. We use a 3$^{rd}$ order polynomial to fit $H_{ex}(P,n)$ with the boundary conditions $H_{ex}(n=0) = 0$ and $H_{ex}(n=1) = 0$ at each pressure:

$$H_{ex}(V,n) = an^3 + bn^2 - (a+b)n \quad (10)$$

which produces

$$\frac{\partial H_{ex}(n)}{\partial n} = 3an^2 + 2bn - (a+b). \quad (11)$$

After obtaining $H_{ex}(V,n)$ (see below), we fit Eq. (10) at each volume and obtain $a(V)$ and $b(V)$. They are as used in Eq. (11) and replace in Eq. (8), resulting in:

$$\Delta G_{LS-HS} + 3an^2 + 2bn - (a+b) + k_B T x_{Fe} \ln\left[\frac{n}{1-n}(m(2S+1))\right] = 0 \quad (12)$$

Equation (12) is then solved numerically for $n$ at each $P,T$. This procedure was followed for $x_{Fe} = 0.1875$.

Next, we obtain $H_{ex}(P,T,n)$:

$$H_{ex}(P,n) = E_{ex}^{st}(V,n) + P_{ex}^{st}(V,n)V \quad (13)$$

where $P_{ex}^{st}(V,n) = -\frac{\partial E_{ex}^{st}}{\partial V}$. The first step in this procedure consisted in obtaining $E_{ex}^{st}(V,n)$.

For 8 different volumes, $E_{ex}^{st}$ was obtained by carrying out ab initio calculations on a 64-atoms supercell with 6 Fe, 26 Mg, and 32 O ions. $n$ varied from 0 to 1, in steps of $\frac{1}{6}$. The supercell Mg/Fe configuration maximized iron-iron distances. The possible HS-LS iron configurations are listed in Table S3. A total of 51 HS-LS configurations are involved but only 10 with different multiplicities are inequivalent.

A typical example of the type of results we produce is seen in Fig. S16.

Fig. S16 shows $E - E_0$ where

$$E = E^{st}(V,T,n) = \frac{1}{N_n} \sum_{i=1}^{N_n} m_i^n E_i(n) e^{-\frac{E_i(n)}{k_B T}}, \quad (14)$$

with $m_i^n$ is the multiplicity of the i$^{th}$ inequivalent configuration with LS-fraction $n$, $N_n = \sum_{i=1}^{N_n} m_i^n$ is the total number of HS-LS configurations

for $n$, $E_0 = E_{HS}(V)$ and $E_{ideal}(V,n) = (1-n)E_{HS}(V) + nE_{LS}(V)$ (blue symbols in Fig. S16). As seen, there is an insignificant temperature dependence in $E^{st}(V,T,n)$ which is rightly disregarded.

Fig. S17 shows $H_{ex}(P,n)$ at different pressures fit to a 3$^{rd}$ order polynomial in $n$ as indicated in Eq. (10).

**Ab initio calculations.** Self-consistent LDA+U$_{sc}$ calculations were performed using the Quantum ESPRESSO code[61]. The projector-augmented wave (PAW) data sets from the PSlibrary[62]. A kinetic-energy cutoff of 100 Ry for wave functions and 600 Ry for spin-charge density and potentials were used. In all cases, atomic orbitals were used to construct occupation matrices and projectors in the LDA+ U$_{sc}$ scheme[63]. The Hubbard parameter $U$ on Fe-3d states was computed using density-functional perturbation theory[64]. A cubic supercell with 64 atoms was constructed, i.e., (Fe$_x$Mg$_{1-x}$)$_{32}$O$_{32}$, with x = 0.1875. The $2 \times 2 \times 2$ **k**-point mesh was used for Brillouin zone integration. Structure optimization was performed by relaxing atomic positions with a force convergence threshold of 0.01 eV/ Å. The convergence threshold of all self-consistent field (SCF) calculations was $1 \times 10^{-9}$ Ry and for DFPT calculations of U$_{sc}$ was $1 \times 10^{-6}$ Ry. Phonon calculations were performed using the finite-displacement method and the Phonopy code[65] with force constants computed with Quantum ESPRESSO. Vibrational density of states (VDOSs) were obtained using a q-point $20 \times 20 \times 20$ mesh. The vibrational contribution to the free energy was calculated using the quasiharmonic approximation with the qha code[66]. More details on these ab initio calculations can be found in[21].

Here we inspect results from four thermodynamic models: a) ideal HS/LS mixing with magnetic entropy effect (MEE) ($G_{mag} = 0$), b) non-ideal mixing with MEE ($G_{mag}$ given by Eq. (6)), c) ideal HS/LS mixing without MEE, b) non-ideal mixing without MEE. The four diagrams for $n(P,T)$ for x = 0.1875 are shown in Fig. S15.

For ideal or non-ideal ISC modeling, the inclusion of MEE decreases the slope of the ISC. With MEE, the crossover pressure range agrees better with data from Komabayashi et al.[67] on a sample with x = 0.19. Without MEE, the slope of the ISC agrees better with Lin et al.[68] data on a sample with $x = 0.25$. This sample showed antiferromagnetic correlations at low temperatures, consistent with Fe-Fe exchange interaction with larger $x$, and lower $S_{mag}$.

The 300 K compression curves for these four models are shown in Fig. S19 below. The inclusion or exclusion of $G_{mag}$ in the calculation is not visible at 300 K for the non-ideal mixing model. MEE is distinguishable in the slope of the ISC only, for ideal or non-ideal solution modeling.

**Thermoelasticity calculations.** The formalism for thermoelasticity with a spin crossover is described in[55]. The components of the compliance tensor in the mixed spin (MS) state are written as:

$$S_{ij}(n)V(n) = nS_{ij}^{LS}V^{LS} + (1-n)S_{ij}^{HS}V^{HS} - \frac{1}{9}\alpha_{ij}\left(V^{LS} - V^{HS}\right)\frac{\partial n}{\partial p}\bigg|_{P,T}. \quad (15)$$

All quantities in Eq. (15) are functions of pressure and temperature, e.g., $V(n) = V(P,T,n)$ or $S_{ij}^{LS} = S_{ij}^{LS}(P,T)$. For this cubic system, $\alpha_{11} = \alpha_{12} = 1$, $\alpha_{44} = 0$. The $S_{ij}^{HS/LS}(P,T)$ are obtained using by inverting the elastic tensor, $C_{ij}^{HS/LS}(P,T)$ calculated with the cij code[69]. The compliance tensor, $S_{ij}(P,T,n)$, is then inverted to give $C_{ij}(P,T,n)$.

Bulk, $K(P,T,n)$, and shear, $G(P,T,n)$, elastic moduli can be determined from the elastic constant $C_{ij}(P,T,n)$ using the Voigt-Reuss-Hill (VRH) averaging scheme. The Voigt average assumes that strain is uniform throughout the system (upper bound). For a polycrystalline system, they are:

$$K_V = \frac{1}{9}\left[(C_{11} + C_{22} + C_{33}) + 2(C_{12} + C_{23} + C_{13})\right] \quad (16a)$$

$$G_V = \frac{1}{15}\left[(C_{11}+C_{22}+C_{33}) - (C_{12}+C_{23}+C_{13}) + 3(C_{44}+C_{55}+C_{66})\right]$$

(16b)

The Reuss bound assumes uniform stress and can be computed as

$$K_R = \frac{1}{\left[(S_{11}+S_{22}+S_{33}) + 2(S_{12}+S_{23}+S_{13})\right]}$$

(17a)

$$G_R = \frac{15}{\left[4(S_{11}+S_{22}+S_{33}) - 4(S_{12}+S_{23}+S_{13}) + 3(S_{44}+S_{55}+S_{66})\right]}$$

(17b)

The arithmetic average of the Voigt and Reuss bounds is the Hill average. Thus, the VRH average of the elastic moduli are

$$K_{VRH} = \frac{K_V + K_R}{2}$$

(18a)

$$G_{VRH} = \frac{G_V + G_R}{2}$$

(18b)

It is implicit that all quantities ($M$) above are functions of pressure, temperature, and $n$, i.e., $M(P,T,n)$, for a particular $x$. Such elastic properties ($M(P,T,x,n)$) were calculated for $x = 0$ and $x = 0.1875$ and then linearly interpolate/extrapolated for $0 \le x \le 0.25$.

The change in $K_{VRH}(P,T,x,n)$, $G_{VRH}(P,T,x,n)$, and the density $\rho(P,T,x,n)$ of the mixed spin state relative to the high spin state for these four models are offered as downloadable files as described under "Data Availability".

## Data availability

Source data for Figs. 1–3, all the thermochemical models in Priors 1, 2, and 3, and the four theoretical models to correct seismic velocities for a spin crossover in ferropericlase, are available at https://doi.org/10.6084/m9.figshare.24328789.

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

## Acknowledgements

LC was supported by a Vidi grant from the Dutch Research Council (NWO) on grant number 016.Vidi.171.022. RMW and JZ acknowledge funding by NSF grant EAR-2000850 and DOE grant DE-SC0019759. This research used resources of the Oak Ridge Leadership Computing Facility, which is a DOE Office of Science User Facility supported under contract DE-AC05-00OR22725 (JT). The authors would like to thank three anonymous reviewers for their helpful comments and insights.

## Author contributions

This work was conceptualized by L.C. and J.Tromp. All authors contributed to the methodology. J.Z. performed the ab initio calculations and contributed to visualization of the results. W.L. prepared the tomography model (GLAD-M25) in the format required for quantitative analysis. L.C. did the thermodynamic modeling of mineral physics data, the fitting of these data to the seismic tomography, and analysis of the results. L.C. prepared the figures together with J.Z.; L.C. wrote the

manuscript together with RMW who described the ab initio calculations in the Methods section. L.C., R.M.W., J.Trampert, and J.Tromp all contributed to reviewing and editing the manuscript.

## Competing interests

The authors declare no competing interests.

## Additional information

**Peer review information** : *Nature Communications* thanks the anonymous, reviewers for their contribution to the peer review of this work. A peer review file is available.

