## [Peer Review File · Nature Communications]

REVIEWER COMMENTS

Reviewer #1 (Remarks to the Author):

Thank you for the opportunity to review the manuscript by Cobden and co-workers.

The manuscript reports a detailed joint comparison of shear and bulk velocities between models exploring a large range of chemical compositions expected in the lower mantle with those derived from a new tomographic model GLAD-25. The results robustly demonstrate a reduced mismatch between models and tomography results if the iron spin crossover in ferropericlase is accounted for. The iron spin crossover strongly effects the physical properties of ferropericlase and the mantle as a whole. As such, the reported results have wide-ranging implications for researchers in geophysics, geodynamics, and geochemistry. I support publication in Nature Communications, but I have a few comments that the authors need to address before acceptance.

Below are my comments:

- In line 35, when discussing joint interpretation of multiple observables, maybe add density as well?
- The paragraph starting at line 38 seem to miss a few key references supporting some of the general statements made, e.g. on the difficulties in joint P- and S—wave interpretations. Could you please add some further citations here?
- Fig. 1, panel D: It seems that the largest remaining difference between the model with ISC and GLAD-M25 remains at depths which are affected by the spin crossover. I wonder if differences in the onset depths and broadness could minimize these (i.e. how would this panel change when using the different model results shown in Fig. S15?)
- Lines 152-155: This sentence could benefit from a reference to Sun et al. (2022) and Fig, S15.
- Line 163: Please add references for PREM and AK135.

- Lines 163-165: Please add a reference to Cammarano et al. (2010), who first looked at the effect of the spin crossover on the velocity gradients with depths.

- Lines 177-185: I think that this part should be within the introduction section of the paper.

- Lines 192-193: Could you please specify which theoretical model results (Fig. S15) were used for Fig. 1 (or are they all included)?

- Line 196: "now enough" sounds weird, maybe reformulate?

- Fig. 2: It seems to me that the theoretical models with non-ideal treatment of the ISC lead to a smaller misfit with GLAD-25 (above D''). Would it be worth discussing this?

- Lines 267-268: I think that Cammarano et al. (2010) should be cited here as they did argue for the spin crossover to change the velocity gradients in 1D seismic profiles, i.e. the statement that a seismic signature was not anticipated seems not quite correct.

- Line 270: Maybe worth saying that a lot of other properties might be affected by the spin crossover, including viscosity, Fe distribution, density.

- Lines 277-278: The conclusion that the signal of the spin crossover is enhanced when using bulk wave speeds is not new and I suggest to cite some early references here (Marquardt et al. 2009, Wu and Wentzcovitch 2014)

- Lines 298-305: The discussion about a possible SiO₂ enrichment in the D'' region feels a bit premature. I suspect that the outcome on this part of the mantle is strongly dependent on the properties of ppv (which are poorly constrained). I would suggest to re-formulate this paragraph, present it as possibility that needs to be explored in the future. Personally, I think that the main point of the paper is the robust detection of the iron spin crossover in the mantle and I would not dilute this message by a much less robust discussion about possible SiO₂ enrichment. I think that the manuscript becomes stronger without this last part, but I leave this to the authors to decide.

- Fig, S16: A very dense experimental V(P) curve has recently been published (Méndez et al. 2022). I suggest to compare the predicted volumes to these new data as they may allow to constrain which theoretical model is most accurate.

In summary, this manuscript presents very interesting findings with important implications for the lower mantle and should be accepted for publication after some revision. I hope that my comments can help to improve the quality and clarity of the manuscript, and balance the discussion of previous works.

Cammarano, F., H. Marquardt, S. Speziale and P. J. Tackley (2010). Role of iron-spin transition in ferropericlase on seismic interpretation: A broad thermochemical transition in the mid mantle? *Geophysical Research Letters* 37: L03308.

Marquardt, H., S. Speziale, H. J. Reichmann, D. J. Frost and F. R. Schilling (2009). Single-crystal elasticity of (Mg_{0.9}Fe_{0.1})O to 81 GPa. *Earth and Planetary Science Letters* 287(3-4): 345-352.

Méndez, A. S. J., S. Stackhouse, V. Trautner, B. Wang, N. Satta, A. Kurnosov, R. Husband, K. Glazyrin, H. P. Liermann and H. Marquardt (2022). Broad elastic softening of (Mg,Fe)O ferropericlase across the iron spin crossover and a mixed-spin lower mantle. *Journal of Geophysical Research: Solid Earth* n/a(n/a): e2021JB023832.

Sun, Y., J. Zhuang and R. M. Wentzcovitch (2022). Thermodynamics of spin crossover in ferropericlase: an improved LDA + Usc calculation. *Electronic Structure* 4(1): 014008.

Wu, Z. and R. M. Wentzcovitch (2014). Spin crossover in ferropericlase and velocity heterogeneities in the lower mantle. *Proceedings of the National Academy of Sciences* 111: 10468–10472.

Reviewer #2 (Remarks to the Author):

The paper by Cobden et al. presents an interdisciplinary investigation testing the hypothesis of iron spin-crossover (ISC) in the lower mantle (LM) at depths of 1000-2500 km using seismic wave speeds as constraints. Their study constructs a wide range of prior temperature and compositional models for the LM and uses *Ab initio* calculations to derive corresponding thermoelastic properties with and without ISC. The results suggest that models with prevalent ISC in the LM provide a significantly better fit to 1° by 1° laterally-averaged bulk (V_b) and shear (V_s) wave speeds within the 1000-2500 km depth range, as resolved by the full-waveform tomographic model of GLAD-M25. Besides, they conclude that silica-enriched composition below 2500 km depth probably indicative of remnant of silicate melts accumulated at the base of the mantle is required to match both V_b and V_s near the core-mantle boundary. The paper clearly outlines their modeling design and approach for testing the ISC hypothesis in the LM and provides sufficiently comprehensive supplementary material to justify the resolution and robustness of V_b and V_s used in the study. Though this study's idea is not original and builds upon

previous research (ref. 21), it presents a nice job on exploring thoroughly all the candidate thermochemical models that agree and disagree with seismic observations, thereby strengthening seismic evidence for the prevalence of ISC in the LM. Furthermore, their work contributes our understanding of the behavior of the minerals in the deep mantle and its implication for the earth's mantle dynamics and evolution. Overall, the paper is commendable in its rigorous approach and interdisciplinary investigation. It would attract the broad, multidisciplinary interests and meet the high standard of the journal. Here are a few points and inquiries I have regarding the paper and require the authors to address them in more detail and provide clarifications for the paper's clarity.

(1) The paper lacks a clear explanation of how the "percent fit" shown in Figure 2 is defined and calculated. Additionally, while the red dashed (ideal ISC with magnetic entropic effect) and blue dashed and solid lines (non-ideal ISC with and without MEE) appear to fit V_b and V_s at 1000-2500 km depth similarly well, it raises a question about the ability of V_b and V_s alone to distinguish which ISC model is more realistic for LM conditions.

(2) Since the "broad" prior candidate models cover all the plausible earth's mantle compositions varying extremely from xenoliths to MORBs, why can none of any models, both with and without ISC, fit V_b and V_s well below 2500 km (especially around 2600-2700 km depth) as noticed in Figure 2? Could there be physical or chemical factors near the core-mantle boundary that were inadequately considered in the thermoelastic modeling and ab initio calculations? A clarification on this discrepancy is needed.

(3) The choice of a 1-s seismic period for anelasticity correction (Table S2) seems at odds with the GLAD-M25 model, which was constrained using full-waveform data with a minimum period of 17 s. This brings into question whether the V_b and V_s obtained from the GLAD-M25 model in Figure 2 were also corrected to the 1-s period.

(4) While the paper argues that GLAD-M25 model constrained from the whole waveform data with the same frequency content result in equally well-resolved V_b and V_s , this may not be rigorously true if considering the phase arrivals and their corresponding finite-frequency sensitivity to V_p and V_s used in the tomographic inversion which could be different even at the same period. Therefore, to validate the equal resolution claim, it's suggested to include a point spread function test for both V_p and V_s , not just for V_s .

(5) Clarification is needed on whether the V_s used in the ISC test represents an average of radially- and transversely-polarized shear wave speeds, given that GLAD-M25 is a transversely isotropic model.

(6) The Methods section lacks explanations for certain physical parameters used in equations, such as Est and F_{vib} in eq. (4), S_{conf} and k_B in eq. (5), S_{mag} in eq. (6), and Hex in eq. (10). To make readers less familiar with the field be easier to follow the theoretical derivation, it is suggested to provide a table similar to Table S2, listing parameter names, definitions, and values used in the modeling.

(7) References or explanations for the software and codes used in the modeling, such as LDA+Usc calculations, Quantum ESPRESSO code, and DFPT calculations, should be provided for clarity.

(8) While the paper highlights the importance of accurate estimates of V_b and V_s from the full-waveform tomographic model of GLAD-M25 that renders their study unique from the other studies, a valuable addition would be comparisons and discussions of the modeled temperature and compositional

gradients in the LM using V_b and V_s estimates from different tomographic models presented in the supplementary material.

Reviewer #3 (Remarks to the Author):

Summary:

The manuscript presents quantitative investigation on iron spin transition (in ferropericlase) in the lower mantle. Differing from a previous attempt (Shephard et al., 2021, which shares one leading author), the current work updated related ab initio calculation results and focused on interpreting a recent developed global full-waveform tomography, GLAD-M25. The authors also used a probabilistic inversion method (Monte Carlo) when constraining the temperature and compositions. They found that fitting model GLAD-M25 using only varying thermal and compositional states gives unsatisfied results. Incorporating spin transition in ferropericlase improves the fit, indicating the presence of such transition at ~ 2200 km. Byproducts from the probabilistic inversion highlights an SiO_2 rich layer above core-mantle boundary, which was hypothesized as magma ocean remnants or subducted MORB materials.

The paper is generally well-written and easy to follow. The findings are of multidiscipline interests including mineralogy, seismology, and geodynamics. I view myself as a seismologist with limited knowledge on mineral physics, so most of my concerns are general or from a seismic perspective. There are two major concerns needed to be addressed.

1. I am wondering if adopted Monte Carlo method well explores the model space. In this current study, there are 6100 composition models and 300 temperature profiles (line 327-334), giving ~ 1.8 million velocity profiles. Considering ISC and/or MEE could multiple the number by a factor of five, approaching ~ 10 million to be compared (e.g., Fig. 2). How many CPU hours are needed to finish the required simulations?

Given the degree of freedom in the compositions (~ 5 in prior 2&3), generating 2500 models seems inadequate. It would be helpful to check if the selected models well cover the model spaces. High-dimensional visualization is hard, so maybe the authors can add a supplementary figure with a few 2D subfigures. Specifically, it will be interesting to see if the abundance of SiO_2 and FeO are correlated. Moreover, from my understanding, prior 1 (pyrolite) is a subset of prior 3 (restricted), and prior 3 is a subset of prior 2 (broad). If so, it would be helpful to see the number of remaining models in prior 2 after removing those satisfying prior 3.

Additionally, I would like to see more details on the Metropolis-Hasting algorithm than figure S3 (line 199). I pasted related main text here. "Where the frequency of the prior is bigger than the target frequency, the algorithm discards a fraction of the mineralogical models in proportion to the relative heights of the bars and using a random number generator to select which particular models are discarded." It is unclear if the discard models are those within a specific "bar" or selected from all existing models. It is also unclear how the authors determine a convergence state, i.e., when to stop the algorithm.

It is unclear how the x-axis values in figure 2 were calculated. Caption in Figure S3 mentioned that the degree of fit was calculated using the two histograms, but I didn't find the details/equations. Please also provide an explanation on why such an approach is preferred and the associated significance levels (statistically).

2. Seismic tomographic models are not perfect. The great similarity between modern global models (such as GAP-P4, SEMUCB-WM1, and the current GLAD-M25) makes me believe large-scale patterns were well-imaged. Meanwhile, as the authors mentioned (line 85-88), absolute seismic velocity at a specific location may have large uncertainties. Therefore, the current manuscript argued that using the distribution of seismic velocity at a given depth is preferred over individual investigation. However, the presented workflow still treats each seismic velocity pair as individual observation, i.e., individual fitting with plausible thermal-compositional models. Consequently, the amplitude of misfit is heavily affected by "extreme" data (Fig. 3). In another word, the majority regions with "explainable" seismic velocity could pass the test without ISC. Therefore, it would be helpful to acknowledge in the main text that the current interpretation may be biased by the adopted tomographic model, and that independent, high-resolution observations are needed to confirm (or against) the absolute seismic velocities in regions where ISC is desperately needed. A supplementary figure showing such a map would be appreciated.

Minor:

Abstract: I would suggest the authors to write a longer abstract. I personally prefer to see an abstract including the motivation, unsolved questions, workflow, and significant results. The unsolved questions and workflow parts were not clear in this current version.

Line 40-42: Despite that the model GLAD-M25 was derived via full-waveform inversion, I would argue it also "utilize only a fraction of the information available in a seismogram" due to limited frequency bands. Therefore, I suggest the author replace this statement with something about "specific data sensitivity" to avoid unnecessary offense.

Line 46: It is unclear why small perturbations could obfuscate a quantitative interpretation. Perhaps you mean underrepresent bias due to damping during linear inversion.

Line 49: As mentioned, “complete” may be improper.

Line 62-63: “equally well-resolved in the mid-mantle”. I didn’t find a similar statement in the GLAD-M25 publication or in its supplement. The cited figures on point spread functions only show results with shear wave perturbations.

Line 66-67: I didn’t see a comparison between observed and reconstructed seismograms. So, I cannot assess if the statement is convincing.

Line 73-75: Mathematically, modeling bulk-shear velocities must be equivalent to fitting P-S wave velocities. Since iron spin transition most affect bulk modulus, such a transition should cause a greater fractional change on bulk velocity than that on P wave velocity, as supported by figure S12. In practice, if the authors account for the uncertainties of P and S wave velocities, the statistical significance of their finding should not be affected by the choice of bulk-shear or P-S pairs. Therefore, it is not surprising they used bulk-shear velocities, but I disagree with the statement on “complicated”.

Line 112: “300,000 pyrolite models”. I think it should be 1100 pyrolite composition model multiply 300 temperatures (supplementary). Maybe there is a typo. I also suggest saying “300,000 models (prior 1)”.

Line 166-168: Please replot the cited figure with a specific mark on where the authors think a change happens.

Line 191-192: Is it possible to deposit the updated velocities to a public domain? It would be appreciated by following studies.

Line 230-232: While the SiO₂ at ~2200 km in these models is abnormally low, I think it is unfair to compare them with reference petrological or cosmochemical compositions. The latter two only offer constraints on a global averaged composition (or at least a large area both in lateral and depth directions). The related temperatures argument is more convincing.

line 344: How many models were discarded?

line 361-363: The details are beyond my expertise, but I am curious about how this inconsistency affect the results.

line 517: Why is here a "(a)"?

Supplementary:

line 164, table S1, typo in Prior 3, "BROAD" should be "RESTRICTED".

REVIEWER COMMENTS

Reviewer #1 (Remarks to the Author):

Thank you for the opportunity to review the manuscript by Cobden and co-workers.

The manuscript reports a detailed joint comparison of shear and bulk velocities between models exploring a large range of chemical compositions expected in the lower mantle with those derived from a new tomographic model GLAD-25. The results robustly demonstrate a reduced mismatch between models and tomography results if the iron spin crossover in ferropericlase is accounted for. The iron spin crossover strongly effects the physical properties of ferropericlase and the mantle as a whole. As such, the reported results have wide-ranging implications for researchers in geophysics, geodynamics, and geochemistry. I support publication in Nature Communications, but I have a few comments that the authors need to address before acceptance.

We very much appreciate this positive feedback from Reviewer 1.

Below are my comments:

- In line 35, when discussing joint interpretation of multiple observables, maybe add density as well?

We have updated the text to include this (now at line 46).

- The paragraph starting at line 38 seem to miss a few key references supporting some of the general statements made, e.g. on the difficulties in joint P- and S—wave interpretations. Could you please add some further citations here?

We have added two key references at line 50 (NB new references do not show up in track changes)

- 4 Robertson, G. S. & Woodhouse, J. H. Ratio of relative S to P velocity heterogeneity in the lower mantle. *Journal of Geophysical Research: Solid Earth* **101**, 20041-20052 (1996). [https://doi.org:https://doi.org/10.1029/96JB01905](https://doi.org/10.1029/96JB01905)
- 5 Ritsema, J. & van Heijst, H.-J. Constraints on the correlation of P- and S-wave velocity heterogeneity in the mantle from P, PP, PPP and PKPab traveltimes. *Geophysical Journal International* **149**, 482-489 (2002). [https://doi.org:10.1046/j.1365-246X.2002.01631.x](https://doi.org/10.1046/j.1365-246X.2002.01631.x)

- Fig. 1, panel D: It seems that the largest remaining difference between the model with ISC and GLAD-M25 remains at depths which are affected by the spin crossover. I wonder if

differences in the onset depths and broadness could minimize these (i.e. how would this panel change when using the different model results shown in Fig. S15?)

We have added the following Figure to the manuscript (Fig. S3) which shows the panel D from Figure 1 (pyrolites with an ISC) plotted with the four different theoretical approximations of ISC. In all cases, the fit is very poor at all depths in the mantle.

For completeness, we also added the following Figure S4, which shows panel E from Figure 1 (restricted, variable composition plus ISC) plotted for the 4 different ISC theoretical approximations:

- Lines 152-155: This sentence could benefit from a reference to Sun et al. (2022) and Fig, S15.

We have added the reference to Sun et al 2022 (ref 21, now at line 173), as well as a reference to Fig S18 (was previously Fig S15).

- Line 163: Please add references for PREM and AK135.

We have added the references (now at line 185). We also added these references at line 101.

- Lines 163-165: Please add a reference to Cammarano et al. (2010), who first looked at the effect of the spin crossover on the velocity gradients with depths.

We have added the reference (reference 24 at line 187).

- Lines 177-185: I think that this part should be within the introduction section of the paper.

We prefer to keep this paragraph in place, as we begin our study investigating whether we can fit seismic wave speeds with simple changes in temperature and composition. When this proves difficult, we then consider the effects of iron spin crossover. If we would move this to the introduction it would imply that the goal of our study was to locate the spin crossover, when this was in fact the serendipitous result. We have however re-ordered the text in the introduction (text previously at lines 93-96 moved to lines 100-103) for improved clarity.

- Lines 192-193: Could you please specify which theoretical model results (Fig. S15) were used for Fig. 1 (or are they all included)? We added an extra sentence to the Figure caption: ***D** and **E** were calculated assuming non-ideal mixing between the high spin (HS) and low spin (LS) states. Plot **D** is without magnetic effects while Plot **E** is with magnetic effects (see Supplementary Information for details).*

- Line 196: “now enough” sounds weird, maybe reformulate?

We changed the phrase to “it becomes possible” (lines 219-220)

- Fig. 2: It seems to me that the theoretical models with non-ideal treatment of the ISC lead to a smaller misfit with GLAD-25 (above D”). Would it be worth discussing this?

The fitting improves by including magnetic entropy effects (MEE) in ideal and non-ideal solid solution (SS) models. Fig. S18 (was Figure S15 in original submission) shows that MEE extends the spin crossover deeper in the mantle, i.e., the fraction of HS states deep in the mantle increases with MEE. Overall, the spin crossover along the mantle geotherm is broader with MEE. Non-ideal SS also broadens the ISC.

Figure R1 shows velocities of an aggregate with 63 vol% of $(\text{Mg}_{0.92}\text{Fe}_{0.08})\text{SiO}_3$ plus 37 vol% of $(\text{Mg}_{0.85}\text{Fe}_{0.15})\text{O}$. Ideal and non-ideal HS/LS solid solutions (SS) of ferropicls are used. The best fit to mantle velocities (shown in Fig.2) comes from the broadest ISC, i.e., the one generated by non-ideal SS with MEE. The fitting also shows that the reduced V_P , V_ϕ , and increased V_S caused by the ISC, irrespective of

the ISC model are essential to improving the fit and producing more reasonable temperatures and compositions.

Figure R1 – Comparison of velocities and densities of an aggregate with 63 vol% of $(Mg_{0.92}Fe_{0.08})SiO_3$ plus 37 vol% of $(Mg_{0.85}Fe_{0.15})O$ for different ISC models along the Brown and Shankland geotherm. (a) ideal HS/LS ferropericalse ISC, with and without MEE, and no ISC. (b) same but for a non-ideal SS model for the ISC.

Change to the manuscript:

- Figure R1 is now included as Fig. S20.
- A statement referring to Fig. S20 is included in lines 252-255.

- Lines 267-268: I think that Cammarano et al. (2010) should be cited here as they did argue for the spin crossover to change the velocity gradients in 1D seismic profiles, i.e. the statement that a seismic signature was not anticipated seems not quite correct. We have added the reference (ref 24; the relevant text is now at line 324).

- Line 270: Maybe worth saying that a lot of other properties might be affected by the spin crossover, including viscosity, Fe distribution, density.

We updated the sentence to mention viscosity (line 326). We did not include Fe distribution and density, because our calculations indicate that the overall density and elastic properties of the bulk assemblage barely change when the iron is re-distributed (please see our response to Reviewer 3's minor comment on lines 361-363 for further details).

- Lines 277-278: The conclusion that the signal of the spin crossover is enhanced when using bulk wave speeds is not new and I suggest to cite some early references here (Marquardt et al. 2009, Wu and Wentzcovitch 2014)

Thanks, we added the references (refs 22 and 32, now line 336).

- Lines 298-305: The discussion about a possible SiO₂ enrichment in the D" region feels a bit premature. I suspect that the outcome on this part of the mantle is strongly dependent on the properties of ppv (which are poorly constrained). I would suggest to re-formulate this paragraph, present it as possibility that needs to be explored in the future. Personally, I think that the main point of the paper is the robust detection of the iron spin crossover in the mantle and I would not dilute this message by a much less robust discussion about possible SiO₂ enrichment. I think that the manuscript becomes stronger without this last part, but I leave this to the authors to decide.

We think the apparent Si enrichment at the bottom of the mantle is at least a highly topical and interesting point of discussion but we agree with Reviewer 1 that the inference of SiO₂ enrichment is less robust than our inference of a spin transition, as several mineralogical phenomena are not included due to lack of constraints on their elastic properties. We added an extra sentence (lines 365-368) to discuss other effects which may be seismically significant but that have not yet been included in calculations.

New text (lines 365-368):

Additionally, ongoing studies in mineral physics will enable the inclusion of the effects of ferric iron³⁹ and calcium-rich bridgmanite⁴², as well as further constraints on the stability and elastic properties of post-perovskite, which may influence the seismic properties in this region and the subsequently modify the inferred chemical composition.

- Fig. S16: A very dense experimental V(P) curve has recently been published (Méndez et al. 2022). I suggest to compare the predicted volumes to these new data as they may allow to constrain which theoretical model is most accurate.

The new Fig. S16 (now Fig. S19) now includes the data from Mendez et al, 2022. Clearly, the non-ideal solution models' compression curves agree better with all experimental compression curves (despite small differences in compositions). However, the difference between the non-ideal models is too subtle and the difference between them at 300 K is smaller than the difference w.r.t. to experiments.

Figure R2 – Compression curves of ferropericlase for different ab initio models and two experiments.

Change to the manuscript: Fig. S19 (was Figure S16 in the original submission) now includes the Mendez 2022 experimental compression curve.

In summary, this manuscript presents very interesting findings with important implications for the lower mantle and should be accepted for publication after some revision. I hope that my comments can help to improve the quality and clarity of the manuscript, and balance the discussion of previous works.

Thank you, we hope we have sufficiently addressed these comments.

Cammarano, F., H. Marquardt, S. Speziale and P. J. Tackley (2010). Role of iron-spin transition in ferropericlase on seismic interpretation: A broad thermochemical transition in the mid mantle? *Geophysical Research Letters* 37: L03308.

Marquardt, H., S. Speziale, H. J. Reichmann, D. J. Frost and F. R. Schilling (2009). Single-crystal elasticity of (Mg_{0.9}Fe_{0.1})O to 81 GPa. *Earth and Planetary Science Letters* 287(3-4): 345-352.

Méndez, A. S. J., S. Stackhouse, V. Trautner, B. Wang, N. Satta, A. Kurnosov, R. Husband, K. Glazyrin, H. P. Liermann and H. Marquardt (2022). Broad elastic softening of (Mg,Fe)O ferropericlase across the iron spin crossover and a mixed-spin lower mantle. *Journal of Geophysical Research: Solid Earth* n/a(n/a): e2021JB023832.

Sun, Y., J. Zhuang and R. M. Wentzcovitch (2022). Thermodynamics of spin crossover in ferropericlase: an improved LDA + Usc calculation. *Electronic Structure* 4(1): 014008.

Wu, Z. and R. M. Wentzcovitch (2014). Spin crossover in ferropericlase and velocity heterogeneities in the lower mantle. *Proceedings of the National Academy of Sciences* 111: 10468–10472.

Reviewer #2 (Remarks to the Author):

The paper by Cobden et al. presents an interdisciplinary investigation testing the hypothesis of iron spin-crossover (ISC) in the lower mantle (LM) at depths of 1000-2500 km using seismic wave speeds as constraints. Their study constructs a wide range of prior temperature and compositional models for the LM and uses Ab initio calculations to derive corresponding thermoelastic properties with and without ISC. The results suggest that models with prevalent ISC in the LM provide a significantly better fit to 1° by 1° laterally-averaged bulk (V_b) and shear (V_s) wave speeds within the 1000-2500 km depth range, as resolved by the full-waveform tomographic model of GLAD-M25. Besides, they conclude that silica-enriched composition below 2500 km depth probably indicative of remnant of silicate melts accumulated at the base of the mantle is required to match both V_b and V_s near the core-mantle boundary. The paper clearly outlines their modeling design and approach for testing the ISC hypothesis in the LM and provides sufficiently comprehensive supplementary material to justify the resolution and robustness of V_b and V_s used in the study. Though this study's idea is not original and builds upon previous research (ref. 21), it presents a nice job on exploring thoroughly all the candidate thermochemical models that agree and disagree with seismic observations, thereby strengthening seismic evidence for the prevalence of ISC in the LM. Furthermore, their work contributes our understanding of the behavior of the minerals in the deep mantle and its implication for the earth's mantle dynamics and evolution. Overall, the paper is commendable in its rigorous approach and interdisciplinary investigation. It would attract the broad, multidisciplinary interests and meet the high standard of the journal. Here are a few points and inquiries I have regarding the paper and require the authors to address them in more detail and provide clarifications for the paper's clarity.

We thank Reviewer 2 for their positive evaluation of our manuscript.

(1) The paper lacks a clear explanation of how the "percent fit" shown in Figure 2 is defined and calculated.

We added an extra sentence to the caption of Figure 2: *Percent fit is defined as the degree of overlap between theoretical wavespeed frequency distributions and observed wavespeed frequency distributions (example frequency distributions are shown in Fig. S6).*

Additionally, while the red dashed (ideal ISC with magnetic entropic effect) and blue dashed and solid lines (non-ideal ISC with and without MEE) appear to fit V_b and V_s at 1000-2500 km depth similarly well, it raises a question about the ability of V_b and V_s alone to distinguish which ISC model is more realistic for LM conditions.

All these ISC models, whether ideal or non-ideal with or without magnetic entropic effects (MEE), improve the fitting of the velocity profiles in the mantle and produce more realistic compositions and temperatures. Analyses of V_b or V_s alone provide fewer constraints on the model.

What is clear is that an ISC is needed, meaning, a change in compressibility and density of ferropericlasite across the ISC is key to providing more realistic thermal-chemical models. The shear modulus is not severely affected by the ISC but the bulk modulus is, resulting in an improved description of bulk and compressive velocities. An analysis of V_s alone is much less affected by which ISC model one uses. It is the ISC itself (its effect on the bulk modulus) that better constrains the amount of ferropericlasite in the deep mantle. Joint V_b - V_s or V_p - V_s analyses better constrain thermal-mineralogical models. Please see Fig. R1 for the effect of the ISC on a typical pyrolytic composition.

Change to the manuscript: No change is needed.

(2) Since the "broad" prior candidate models cover all the plausible earth's mantle compositions varying extremely from xenoliths to MORBs, why can none of any models, both with and without ISC, fit V_b and V_s well below 2500 km (especially around 2600-2700 km depth) as noticed in Figure 2? Could there be physical or chemical factors near the core-mantle boundary that were inadequately considered in the thermoelastic modeling and ab initio calculations? A clarification on this discrepancy is needed.

Figure 2 shows the fit to Prior 3 (this is already stated in the Figure caption), which is the "restricted" chemical variations. Figure S7 (was Figure S4 in the original submission) shows the results for the "broad" prior (Prior 2) and in that case, it is indeed possible to fit the seismic tomography to > 99% at all depths. The misfit of Prior 2 below 2500 km can be explained by the lack of SiO₂-rich models (this is already discussed in the main text at lines 289-293).

(3) The choice of a 1-s seismic period for anelasticity correction (Table S2) seems at odds with the GLAD-M25 model, which was constrained using full-waveform data with a minimum period of 17 s. This brings into question whether the V_b and V_s obtained from the GLAD-M25 model in Figure 2 were also corrected to the 1-s period.

We added an extra sentence to the Methods to clarify this (lines 451-453): *The anelasticity correction is performed at a period of 1 s, as we work with a version of GLAD-M25 in which the wave speeds have been shifted to 1 second period.*

(4) While the paper argues that GLAD-M25 model constrained from the whole waveform data with the same frequency content result in equally well-resolved V_b and V_s , this may not be rigorously true if considering the phase arrivals and their corresponding finite-frequency sensitivity to V_p and V_s used in the tomographic inversion which could be different even at the same period. Therefore, to validate the equal resolution claim, it's suggested to include a point spread function test for both V_p and V_s , not just for V_s .

We appreciate the reviewer's suggestion. While Lei et al. 2020 (the GLAD-M25 paper) contains many point-spread functions, it is an omission on the part of the authors that no PSF was calculated for P-waves. Unfortunately calculating a point spread function comes at a considerable computational cost of one full iteration, and our current INCITE (US DOE) allocation is for the construction of GLAD-M35, not for further assessments of GLAD-M25. (We've been trying elsewhere, but it's very slow going.) GLAD-M35 will be a joint P & S model with proper uncertainty quantification based on random probing of the Hessian, and we believe this is the way to do uncertainty quantification in full-waveform inversion going forward. We have also been working on a preliminary thermochemical interpretation of GLAD-M35 and there are only very minor differences between M25 and M35, which gives us confidence that our interpretation of GLAD-M25 is robust.

(5) Clarification is needed on whether the V_s used in the ISC test represents an average of radially- and transversely-polarized shear wave speeds, given that GLAD-M25 is a transversely isotropic model.

We added the word "isotropic" to line 412 in the Methods as follows:

*For each of our thermochemical models, corresponding phase relations and **isotropic** elastic properties are calculated using Perple_X thermodynamic modelling software⁵⁰ together with the elastic parameters, equation of state, and solid solution model of^{51,52}*

(6) The Methods section lacks explanations for certain physical parameters used in equations, such as E_{st} and F_{vib} in eq. (4), S_{conf} and k_B in eq. (5), S_{mag} in eq. (6), and H_{ex} in eq. (10). To make readers less familiar with the field be easier to follow the theoretical derivation, it is suggested to provide a table similar to Table S2, listing parameter names, definitions, and values used in the modeling.

All these quantities are well-defined now.

Change to the manuscript:

- a) We included definitions of E_{st} and F_{vib} in lines 485-487.
- b) We included definitions of S_{conf} and k_B in lines 495-503.
- c) We included the definition of H_{ex} and detailed its contributions in lines 532-539. These quantities were defined in the previous text but not at the point the referee wanted. Now they are in a more suitable place. No further changes are needed.

(7) References or explanations for the software and codes used in the modeling, such as

LDA+Usc calculations, Quantum ESPRESSO code, and DFPT calculations, should be provided for clarity.

We have included such references now.

Change to the manuscript:

a) References 59, 60, and 62 are now included.

(8) While the paper highlights the importance of accurate estimates of V_b and V_s from the full-waveform tomographic model of GLAD-M25 that renders their study unique from the other studies, a valuable addition would be comparisons and discussions of the modeled temperature and compositional gradients in the LM using V_b and V_s estimates from different tomographic models presented in the supplementary material.

Our reason for working with GLAD-M25 is that it provides consistent V_b and V_s models as well as absolute wave speeds, which makes a joint interpretation feasible. There are limited alternatives for combined studies of V_b and V_s , and no alternative for working with absolute wave speeds. We have expanded the abstract and added extra references (as per the requests of Reviewers 3 and 1) to emphasise this point more strongly, so that it is clearer why our method cannot easily be replicated with another tomography model. Additionally, performing a thermochemical interpretation of a single tomography model is itself a major undertaking requiring months of work, and repeating the analysis with multiple tomography models would be beyond the scope of this study.

Reviewer #3 (Remarks to the Author):

Summary:

The manuscript presents quantitative investigation on iron spin transition (in ferropericlase) in the lower mantle. Differing from a previous attempt (Shephard et al., 2021, which shares one leading author), the current work updated related ab initio calculation results and focused on interpreting a recent developed global full-waveform tomography, GLAD-M25. The authors also used a probabilistic inversion method (Monte Carlo) when constraining the temperature and compositions. They found that fitting model GLAD-M25 using only varying thermal and compositional states gives unsatisfied results. Incorporating spin transition in ferropericlase improves the fit, indicating the presence of such transition at ~2200 km. Byproducts from the probabilistic inversion highlights an SiO₂ rich layer above core-mantle boundary, which was hypothesized as magma ocean remnants or subducted MORB materials.

The paper is generally well-written and easy to follow. The findings are of multidiscipline interests including mineralogy, seismology, and geodynamics. I view myself as a seismologist with limited knowledge on mineral physics, so most of my concerns are general or from a seismic perspective. There are two major concerns needed to be addressed.

We thank Reviewer 3 for their positive evaluation.

1. I am wondering if adopted Monte Carlo method well explores the model space. In this current study, there are 6100 composition models and 300 temperature profiles (line 327-334), giving ~1.8 million velocity profiles. Considering ISC and/or MEE could multiple the

number by a factor of five, approaching ~10 million to be compared (e.g., Fig. 2). How many CPU hours are needed to finish the required simulations?

We have finished all the required simulations. For each composition model it takes ~1 hour on one CPU to calculate the phase relations and elastic properties for 300 different temperatures at 19 different pressures. It thus took several weeks to run the simulations as I (LC) ran them on my desktop machine, but they were all completed when we wrote the paper. We did not plot every model in the paper, in the interests of clarity and conciseness.

The correction to the velocities for iron spin crossover, of which there are 4 variants (+/- MEE, ideal/non-ideal solid solution) is very fast and simply involves a 2D interpolation of data files which prescribe the change in elastic properties (bulk modulus, shear modulus, density) as a function of 1. temperature and 2. fraction of Fe in ferropericlasite (at a given pressure).

To answer the reviewer's query about whether the method explores the model space sufficiently: For any given Prior, the seismic properties manifest as a cloud in V_b, V_s space (see Figures 1 and S3, S4). The ranges of the clouds depend on the width of the Prior. Increasing the number of models increases the density of points within the cloud, without expanding the ranges. This is illustrated in Figure S2 (a new figure in response to the Reviewer 3's next request): Priors 2 and 3 contain the same number of models (~750 k), but Prior3 covers a narrower range of composition space. This means that it samples the compositional space more densely than Prior 2. Hence, in Figure S2, the density of points is higher for Prior 3 plots than Prior 2 plots.

Updates to the manuscript:

We hope that the addition of Figures S3 and S4 makes it clearer that the simulations are complete. Furthermore, the velocity profiles to which the reviewer refers have now been uploaded onto Figshare (DOI <https://doi.org/10.6084/m9.figshare.24328789>).

We have also uploaded the data tables which quantify the change in elastic properties as a function of P, T and x_{Fe} for the 4 theoretical approximations onto Figshare (same DOI).

Please see our response to Reviewer 3's next comment for additional relevant updates.

Given the degree of freedom in the compositions (~5 in prior 2&3), generating 2500 models seems inadequate. It would be helpful to check if the selected models well cover the model spaces. High-dimensional visualization is hard, so maybe the authors can add a supplementary figure with a few 2D subfigures. Specifically, it will be interesting to see if the abundance of SiO₂ and FeO are correlated. Moreover, from my understanding, prior 1 (pyrolite) is a subset of prior 3 (restricted), and prior 3 is a subset of prior 2 (broad). If so, it would be helpful to see the number of remaining models in prior 2 after removing those satisfying prior 3.

As per the reviewer's request, we prepared supplementary Figure S2 which shows scatter plots of the Fe-Mg-Si weight percentages in Priors 2 and 3. There is no significant correlation between Fe and Si, nor between Fe and Mg. There is a degree of anti-correlation between Si and Mg (especially for Prior 2) as these are the most abundant components, and since the total wt% of all the oxides must equal 100%, then necessarily both components cannot be simultaneously large or simultaneously small (e.g. it is not possible to have a model with 70 wt% SiO₂ and 50 wt% MgO, even if their priors cover these ranges, because this would produce a total of 120%).

Figure S2

Prior 3 is constructed independently of Prior 2. Since Prior 3 encompasses a narrower range of compositions than Prior 2, but both priors contain the same number of models (~750 k), this means that Prior 3 samples the compositional space more densely. This is visible in Figure S2, where you can see that the density of points is higher in Prior 3 than in Prior 2. Figure S2, together with Figure 1 thus illustrates another important point: that it is the **ranges of the priors** which determine the diversity of seismic and thermochemical properties, while the **number**

of models determines how densely those ranges are sampled. The number of models used in our simulations was in turn determined by the Metropolis-Hastings algorithm, as this always discards a percentage of the models at each iteration, therefore it is important to have sufficient models for the algorithm to converge, but more models are not required.

Furthermore, when the Prior distribution has no overlap with a substantial portion of the Target distribution, the MH also breaks down. This is the case with Prior 1, the pyrolite models (i.e. substantial portions of the black clouds do not overlap with the coloured clouds in Figure S3).

In addition to adding Figure S2, we modified the text in two sections of the Methods: Lines 387-390:

Prior to normalization, models are selected independently for each oxide, but normalization inevitably introduces a degree of anti-correlation between the most abundant oxides (MgO and SiO₂) since the total amount cannot exceed 100%. The other components are uncorrelated (Figure S2).

Lines 405-409:

The ranges of the priors determine the ranges of seismic properties (Figure 1, Figure S3, Figure S4), while the number of models per Prior determines how densely those ranges are sampled (e.g. compare the density of points for Priors 2 and 3 in Figure S2). It is important that there are sufficient models available for the Metropolis-Hastings algorithm to converge, but more models are not necessary.

In the Methods section “Building the prior” we added a couple of extra “each”s to the text to make it clear that each of Priors 2 and 3 contains 750k models and each was constructed independently.

Additionally, I would like to see more details on the Metropolis-Hasting algorithm than figure S3 (line 199). I pasted related main text here. “Where the frequency of the prior is bigger than the target frequency, the algorithm discards a fraction of the mineralogical models in proportion to the relative heights of the bars and using a random number generator to select which particular models are discarded.” It is unclear if the discard models are those within a specific “bar” or selected from all existing models. It is also unclear how the authors determine a convergence state, i.e., when to stop the algorithm.

We have updated the text in lines 227-235 to describe this process more clearly as follows (new text is indicated in bold):

The algorithm considers each bar of the histogram in turn. Where the frequency of the prior is bigger than the target frequency for that bar, the algorithm discards a fraction of the mineralogical models in proportion to the relative heights of the bars and using a random number generator to select which particular models are discarded. Where the frequency of the prior is less than the target frequency, all models are retained. New histograms are constructed for the retained

models, and this procedure is iterated until the best fit between the retained models and the target is obtained. With $\sim 10^6$ models in the prior, the algorithm usually converges within 10-20 iterations, but the exact number depends on how similar the two histograms are initially.

It is unclear how the x-axis values in figure 2 were calculated. Caption in Figure S3 mentioned that the degree of fit was calculated using the two histograms, but I didn't find the details/equations. Please also provide an explanation on why such an approach is preferred and the associated significance levels (statistically).

We added the following text at lines 237-245:

“We quantify the fit of a given set of retained models (the posterior) to the target distribution (V_i or V_s of GLAD-M25) as follows: For the i 'th bar of the histogram, we define f_{min} as whichever is the lower value out of: the frequency of the posterior (f_p) and the frequency of the target (f_t). Since both distributions are normalised to 100%, the fit is then given by the sum of f_{min} over all values of i . In other words:

$$fit (\%) = \sum mm (f_i, f^t)$$

#

We use this approach because it is simple and illustrates the degree of fit clearly.”

2. Seismic tomographic models are not perfect. The great similarity between modern global models (such as GAP-P4, SEMUCB-WM1, and the current GLAD-M25) makes me believe large-scale patterns were well-imaged. Meanwhile, as the authors mentioned (line 85-88), absolute seismic velocity at a specific location may have large uncertainties. Therefore, the current manuscript argued that using the distribution of seismic velocity at a given depth is preferred over individual investigation. However, the presented workflow still treats each seismic velocity pair as individual observation, i.e., individual fitting with plausible thermal-compositional models. Consequently, the amplitude of misfit is heavily affected by “extreme” data (Fig. 3). In another word, the majority regions with “explainable” seismic velocity could pass the test without ISC. Therefore, it would be helpful to acknowledge in the main text that the current interpretation may be biased by the adopted tomographic model, and that independent, high-resolution observations are needed to confirm (or against) the absolute seismic velocities in regions where ISC is desperately needed. A supplementary figure showing such a map would be appreciated.

Our method has the advantage that it is precisely **not** heavily affected by extreme values. The extreme values lie in the tails of the histograms. Since we are fitting histograms rather than individual compositional models, then a misfit in the tails of the histograms becomes negligible relative to the fit of the bulk of the distribution which does not lie in the tails.

It is already visible in Figure 1C that the misfit between thermochemical models without an ISC and seismic tomography covers a substantial portion of the wave speed distribution (look especially at the (non)overlap of the coloured and black clouds at 1900 and 2200 km), and does not lie in the tails.

The advantage of working with absolute velocities is that it allows us to constrain absolute temperatures and compositions, which have been one of the key clues to the presence of the ISC (the fact that models without an ISC are both cold and extremely Si-depleted would not be visible when looking at perturbations). We have

added an extra sentence in the final “Implications” section to better emphasise this (lines 331-333):

By modelling absolute wave speeds (rather than perturbations from a 1-D average), we can constrain absolute temperatures and compositions, which in turn allows us to distinguish which scenarios are (im)plausible.

Minor:

Abstract: I would suggest the authors to write a longer abstract. I personally prefer to see an abstract including the motivation, unsolved questions, workflow, and significant results. The unsolved questions and workflow parts were not clear in this current version.

We have extended the abstract to better explain the motivation and unsolved questions.

Line 40-42: Despite that the model GLAD-M25 was derived via full-waveform inversion, I would argue it also “utilize only a fraction of the information available in a seismogram” due to limited frequency bands. Therefore, I suggest the author replace this statement with something about “specific data sensitivity” to avoid unnecessary offense.

GLAD-M25 uses anything and everything in the 17 to 150 s frequency band. This is definitely not a fraction of the information available in a seismogram. We modified the text (line 74) to make this point clearer. Modification is shown in bold:

*GLAD-M25 constrains bulk and shear wave speeds simultaneously using the same data and over the same range of seismic frequencies (**everything in the range 17 s to 150 s**)*

Line 46: It is unclear why small perturbations could obfuscate a quantitative interpretation. Perhaps you mean underrepresent bias due to damping during linear inversion.

We extended the sentence (now at lines 56-59) as follows (modifications shown in bold)

*This further obfuscates a quantitative interpretation **because many different combinations of thermal or compositional changes – as well as errors – can produce the observed signals. Furthermore, damping and regularization underestimate the true amplitudes of wave speed variations.***

Line 49: As mentioned, “complete” may be improper.

We deleted the word “complete” (now at line 62)

Line 62-63: “equally well-resolved in the mid-mantle”. I didn’t find a similar statement in the GLAD-M25 publication or in its supplement. The cited figures on point spread functions only show results with shear wave perturbations.

Please see our response to point 4 of Reviewer 2 above.

Line 66-67: I didn’t see a comparison between observed and reconstructed seismograms. So, I cannot assess if the statement is convincing.

GLAD-M25 is built using 20 million seismograms, making it impossible to demonstrate a comparison for all the seismograms that went into it. We refer the reviewer to Table 1 in Lei et al. 2020, which assesses the variance reduction in 4

inversion categories on 3 components. Figure 5 shows the behaviour of the misfit function. These are the tools we can use to assess improvements in fit in the inversion.

Line 73-75: Mathematically, modeling bulk-shear velocities must be equivalent to fitting P-S wave velocities. Since iron spin transition most affect bulk modulus, such a transition should cause a greater fractional change on bulk velocity than that on P wave velocity, as supported by figure S12. In practice, if the authors account for the uncertainties of P and S wave velocities, the statistical significance of their finding should not be affected by the choice of bulk-shear or P-S pairs. Therefore, it is not surprising they used bulk-shear velocities, but I disagree with the statement on “complicated”.

We changed complicated to “challenging” (now at line 88): it is more difficult to interpret a smaller signal that is influenced by three parameters (K, G, rho) than one which is influenced by two parameters (K and rho).

Line 112: “300,000 pyrolite models”. I think it should be 1100 pyrolite composition model multiply 300 temperatures (supplementary). Maybe there is a typo. I also suggest saying “300,000 models (prior 1)”.

Thanks for pointing this out. It is indeed a typo and should say 330,00 models. We have corrected the text in the Figure 1 caption, as well as clarifying which sub plots are for Prior1, Prior 2 and Prior 3.

Line 166-168: Please replot the cited figure with a specific mark on where the authors think a change happens.

We prefer not to (over)interpret Figures from other studies without performing a fully quantitative analysis of the data. We invite readers to make their own interpretation.

Line 191-192: Is it possible to deposit the updated velocities to a public domain? It would be appreciated by following studies.

We have uploaded all the details of the three Priors onto Figshare as part of the resubmission (DOI <https://doi.org/10.6084/m9.figshare.24328789>), together with the four theoretical corrections for iron spin crossover. We referenced this DOI in the section “Data Availability” at lines 670-673.

Line 230-232: While the SiO₂ at ~2200 km in these models is abnormally low, I think it is unfair to compare them with reference petrological or cosmochemical compositions. The latter two only offer constraints on a global averaged composition (or at least a large area both in lateral and depth directions). The related temperatures argument is more convincing. We see a major excursion in the SiO₂ values (>5% lower than pyrolite/peridotite) over a depth range of > 1000 km and think that this would be large enough to influence the global average. To emphasise this point we modified the text in lines 280-282 as follows (changes highlighted in bold):

The values of SiO₂ (< 36 wt%) are much lower than any model proposed for the bulk mantle composition on the basis of petrological or cosmochemical arguments^{14,17,29-31} and extend over a depth range of several hundred kilometres, which is volumetrically significant when integrated over the whole mantle.

line 344: How many models were discarded?

We have added Table S2 which shows the number of remaining models per Prior and per depth once the thermodynamically unstable models were removed. Table S2 is referenced in the Methods at lines 416-417 as follows:

The number of remaining models after thermodynamically unstable ones have been discarded is shown in Table S2.

line 361-363: The details are beyond my expertise, but I am curious about how this inconsistency affect the results.

This is an important question. We have results including iron partitioning but we are not ready to include them in this paper. We can confirm that iron partitioning between Fp and Bm will change the composition and velocities of both phases separately but not of the aggregate density or velocities significantly. The most affected velocities are Vb and Vp of the aggregate but the effect of iron partitioning is really minor compared to that of the ISC alone, i.e., without iron partitioning. The effect on Vs is similar but much smaller. The detailed analysis of the effect of iron partitioning is beyond the scope of this work. Nevertheless, on the basis of Birch's law, we could anticipate that the inclusion of iron partitioning will not affect the present conclusions in any significant way. A long manuscript is in preparation on this topic.

Change to the manuscript:

We have included the following statement in lines 435-441 of the manuscript:

Although we do not include the effect of iron partitioning across the ISC on aggregates' densities and velocities, on the basis of Birch's law, we anticipate that such effect will be relatively small. Birch's law relates the compressive velocity with the aggregate density, which depends primarily on the unchangeable aggregate composition. This topic is beyond the scope of this work and we anticipate that our conclusions will not be significantly affected by the iron partitioning effect.

line 517: Why is here a "(a)"?

Good question. We have deleted it.

Supplementary:

line 164, table S1, typo in Prior 3, "BROAD" should be "RESTRICTED".

Thanks for spotting this; we fixed it.

REVIEWERS' COMMENTS

Reviewer #1 (Remarks to the Author):

Dear authors,

Thank you for taking my comments into account in revising your manuscript. I feel that my concerns have been adequately addressed and I am looking forward to publication of the revised paper. Congratulations on this nice paper!

I only have one last, but important request. Could you please add citation to the paper below, which has been published earlier this year, reporting very similar findings. I think that mentioning this paper is indispensable for reasons of good scientific practice and transparency.

Trautner, V. E., et al (2023). "Compressibility of ferropericlase at high-temperature: Evidence for the iron spin crossover in seismic tomography." *Earth and Planetary Science Letters* 618: 118296.

Reviewer #2 (Remarks to the Author):

The authors have made substantial effort in addressing the comments and suggestions from all three reviewers, significantly enhancing the manuscript, figures, tables, and supplementary material. They also provide comprehensive point-by-point responses which clarify the unclear points in the manuscript and improve the overall quality of their study and paper.

One notable improvement is the clarity provided in the Methodology section, specifically addressing concerns raised by the third reviewer regarding the application of the MCMC method. The authors have concisely and clearly explained how this method explores the model space, leading to the convergence of the "target" model distribution that fits seismic velocity observations. These additional details increase the reproducibility and reliability of their results.

While there remain a few points, such as the demonstration of GLAD-M25's Vp model resolution compatibility with the Vs model and testing against other tomographic models, the authors' response is reasonable. The rationale provided, emphasizing the unique strengths of the GLAD-M25 model based on the inversion of full-waveform data for Vp and Vs simultaneously, is convincing. The acknowledgment of the limitations of the study is more transparent in the revised manuscript, providing a clear assessment of the scope and avenues for future research.

I agree with the authors' perspective that, despite the remaining points, the GLAD-M25 model, offering superior constraints on absolute Vp and Vs compared to other travelttime-based tomographic models, is a robust contribution to their study of the thermochemical properties of the earth's deep interior. Based on the paper's enhanced clarity and positive improvements, I agree the acceptance of the paper for publications in Nature Communications.

Reviewer #3 (Remarks to the Author):

I am satisfied with the changes in the current manuscript and recommend its publication on Nature Communications after a few minor text editions. Line numbers are from the clear version.

1. Some SiO₂(subscript) are "SiO2".

2. Line 71. In authors' reply to question 4 from reviewer 2, they mentioned a point spread function of Vp in GLAD-M25 is challenging. Therefore, it would be safer to delete the word "equally" in this sentence.

3. Line 120. An extra "0" in "330,000 pyrolite models".

4. Line 582. The strange "(a)" is still there.

Personal note:

Line 46. In term of my comment on "a fraction of the information available in a seismogram", I personally consider "fraction" as an opposite of "complete or 100%". The construction of GLAD-M25 utilized "everything between 17 s – 150 s", but modern seismometers can record a broader frequency band than that. Higher frequency data also contributed to the understanding of Earth's interiors. Taking the 410/660 discontinuities as examples, seismograms must contain information of the topography variation of these sharp mantle discontinuities, but the GLAD-M25 chosen to ignore them for now (part 4 in Lei et al., 2020, GJI).

I am not criticizing full-waveform tomography and I do believe the technique represents the future direction of seismic imaging. I simply think it would be better to emphasize the limitation of our seismic models.

We would like once again to thank all three reviewers for their positive evaluations and helpful comments on our revised manuscript. We really appreciate this.

We have addressed their final comments as follows:

- We have included the reference to Trautner et al (2023) at line 282
- We adjusted 6 instances of SiO₂ to SiO₂
- We deleted the word “equally” at line 71
- We deleted the extra “0” in 330,000 in the caption for Figure 1
- We deleted the rogue “(a)” in the Methods section

We noted reviewer 3’s personal note and respect their point but would respectfully disagree with it. We consider “fraction” to be bigger than zero but generally less than 50%.

Thanks and kind regards,

Laura Cobden, on behalf of all authors.

Utrecht, 11 December 2023